# Triple Preference Optimization: Achieving Better Alignment using a Single Step Optimization

**Amir Saeidi**                                             *ssaeidi1@asu.edu*
*School of Computing and Augmented Intelligence*
*Arizona State University*

**Shivanshu Verma**                                        *sverma76@asu.edu*
*School of Computing and Augmented Intelligence*
*Arizona State University*

**Kashif Rasul**                               *rasul.kashif@morganstanley.com*
*Morgan Stanley*
*New York, NY*

**Aswin RRV**                                               *aravik13@asu.edu*
*School of Computing and Augmented Intelligence*
*Arizona State University*

**Chitta Baral**                                              *chitta@asu.edu*
*School of Computing and Augmented Intelligence*
*Arizona State University*

**Reviewed on OpenReview:** *https://openreview.net/forum?id=A4jyaZheE8*

## Abstract

Reinforcement Learning with Human Feedback (RLHF) enhances the alignment of Large Language Models (LLMs). However, its limitations have led to the development of Direct Preference Optimization (DPO), an RL-free approach designed to overcome these shortcomings. While studies have shown that DPO improves instruction-following capabilities, it negatively impacts the reasoning ability of LLMs. Additionally, DPO is highly sensitive to judgment noise in preference datasets and the size of the training set. Although several modifications to DPO have been proposed, they still fail to fully resolve these issues. To address these limitations, we propose **Triple Preference Optimization (TPO)**, a new preference learning method designed to enhance both reasoning and instruction-following abilities through one-step optimization. We compare TPO against DPO and its recent variants using state-of-the-art training setups, including both base and instruction-tuned models such as Mistral and Llama 3. Our evaluation covers a comprehensive range of chat-based and reasoning benchmarks. The results demonstrate that TPO achieves significant improvements over existing methods without substantially increasing response length across different dataset sizes. Specifically, TPO outperforms DPO and SimPO by up to **7.0%** and **7.3%** points on Arena-Hard, **12.2%** and **13.3%** points on MixEval-Hard, **10.4%** and **10.1%** points on MMLU-Pro, and **19.0%** and **19.2%** points on GSM8K, respectively. Furthermore, TPO achieves these improvements while requiring less data than DPO.

## 1 Introduction

Since InstructGPT, most Large Language Models (LLMs) add preference optimization on top of pre-training and supervised fine-tuning (Ouyang et al., 2022b). InstructGPT used Proximal Policy Optimization (PPO) (Schulman et al., 2017) as a form of Reinforcement Learning with Human Feedback (RLHF) (Stiennon et al., 2020) for preference

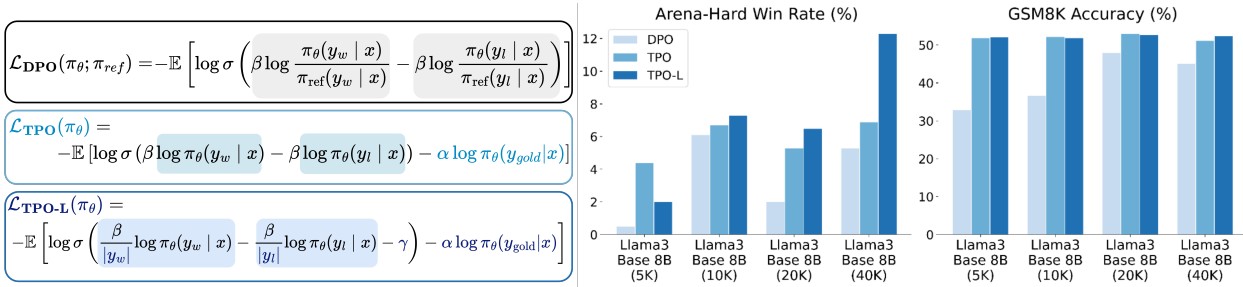

Figure 1: TPO and TPO-L differ by removing the reference model and adding a Behavioral Cloning objective with a regularization term for gold preferences, distinct from preferred and rejected responses. TPO and TPO-L outperform DPO in instruction following and reasoning benchmarks simultaneously.

optimization, and used the Bradley-Terry modeling to build an explicit reward model that is derived from the preference data. Direct Preference Optimization (DPO) (Rafailov et al., 2024b) was proposed as an RL-free alternative to preference learning and optimizes a policy model using an implicit reward function where the regular KL divergence acts as the implicit reward.

Recent studies (Tunstall et al., 2023; Xu et al., 2024) indicate that DPO, which uses multi-step optimization, faces challenges related to optimization inefficiency. Additionally, a new finding (Meng et al., 2024) suggests that KL divergence, which is used in DPO, may not effectively represent a reward. Instead, the average likelihood of a response given the input appears to be a more suitable approach for reward modeling. Motivated by these issues, several variants of DPO have been proposed (Azar et al., 2023; Ethayarajh et al., 2024; Hong et al., 2024b). As highlighted in (Wu et al., 2024; Meng et al., 2024), the above-mentioned DPO variants enhance performance in instruction-following tasks; however, they also point out that there is a notable decline in performance on reasoning benchmarks such as GSM8K. Furthermore, we show that most current preference optimization methods are still sensitive to training set size.

Motivated by the desire to address the challenges faced by the current methods and to simplify the optimization part to a single step, we propose **Triple Preference Optimization (TPO)**, a novel preference optimization algorithm designed to achieve superior performance on both instruction-following and reasoning benchmarks in a single-step optimization process. TPO optimizes a pre-trained model by maximizing the likelihood of gold response ($y_\text{gold}$) using a Behavioral Cloning (BC) objective while incorporating a preference optimization term in a reference-free format. This term increases the likelihood of preferred responses ($y_w$) and decreases the likelihood of rejected responses ($y_l$). Moreover, to enhance control over response length, inspired by the SimPO (Meng et al., 2024) method, we introduce **TPO-L**, a length-controlled variant of TPO. In TPO-L, the average length of preferred and rejected responses constrains the policy model, ensuring that it generates longer responses only when quality improves.

We provide a theoretical formulation of TPO by deriving the optimization function of TPO from Maximum Entropy Reinforcement Learning (MERL) by considering a BC objective on gold response as an entropy. We experimentally demonstrate that TPO addresses the optimization conflict challenge present in some recent preference optimization methods. Additionally, we show that using the average likelihood of a sequence of responses provides a more effective representation of the implicit reward than the definitions used in DPO and SimPO. Comprehensive experiments demonstrate that TPO and TPO-L achieve simultaneous improvements in instruction-following and reasoning benchmarks, demonstrating greater robustness across varying training dataset sizes. Furthermore, we observe that TPO exhibits stronger resilience to judgment noise in preference datasets compared to DPO. Notably, even with half the amount of data, TPO and TPO-L consistently outperform DPO.

In summary, our contributions are as follows:

• We propose Triple Preference Optimization (TPO) and TPO-L, two novel single-step preference learning methods that achieve significantly better performance on smaller training datasets compared to existing methods. Extensive experiments demonstrate that TPO and TPO-L outperform current preference optimization methods on both instruction-following and reasoning benchmarks across various training dataset sizes (See Tables 2 and 3).

• We derive and justify the TPO loss function using Maximum Entropy Reinforcement Learning (MERL), Bradley-Terry modeling, and a variation of the reward function formulation in DPO. Additionally, we demonstrate that $\log \pi_\theta(y|x)$ serves as an implicit reward in TPO, improving reward modeling compared to DPO and SimPO (see Figures 5 and 8).

• We also illustrate that TPO overcomes the optimization conflict challenge in current methods (See Figure 4), exhibits greater robustness to judgment noise in data (See Figure 7), and achieves comparable performance while utilizing less data then DPO (See Table 5).

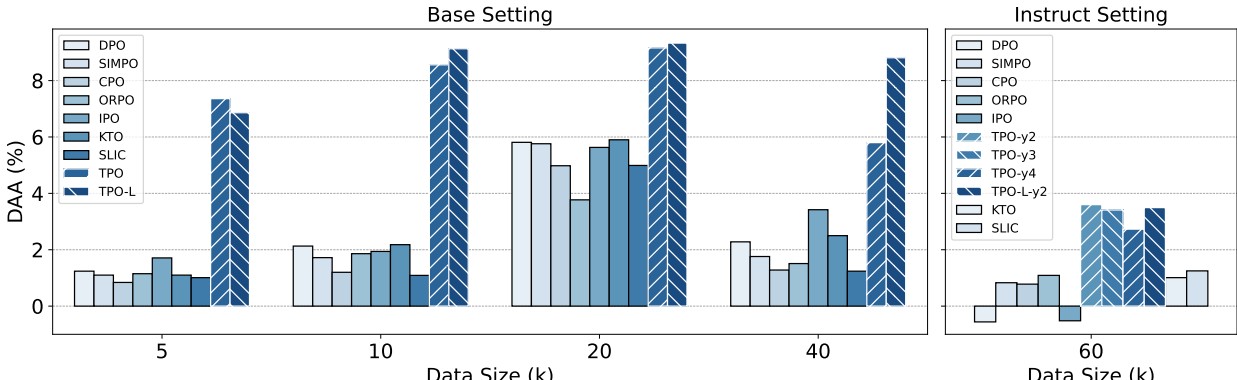

Figure 2: Comparison of post-training improvements over the SFT checkpoint across different methods, measured by the DAA metric.

## 2 Method

In this section, we start by discussing the ongoing challenges in current Preference Optimization (PO) methods, which motivates the development of our new method. We then introduce the Triple Preference Optimization (TPO) method, a novel preference learning algorithm, and provide a detailed explanation of the theoretical foundations supporting it.

### 2.1 Challenges

Current PO methods face several key challenges. First, their *generalization across benchmarks* is limited, as improvements observed in instruction-following tasks often do not carry over to downstream tasks, with some methods performing worse than their Supervised Fine-Tuning (SFT) checkpoints (Meng et al., 2024). To address this, we introduce Difference Average Accuracy (DAA), an effective metric for evaluating post-training improvement.

$$\text{DAA (\%)} = \Delta(\text{Accuracy}_{\text{SFT}}, \text{Accuracy}_{\text{Preference Method}}),$$

where $\text{Accuracy}_{\text{SFT}}$ and $\text{Accuracy}_{\text{Preference Method}}$ represent the average accuracy across five downstream tasks on the SFT checkpoint and the post-training checkpoint, respectively. Additionally, these PO methods are highly sensitive to dataset sizes (see Table 2), which poses challenges for some tasks like reasoning, where data collection is costly. Another major limitation is the reliance on *multiple-step optimization*, where methods like DPO require sequential optimization stages, and attempts to bypass SFT, such as in ORPO, have yet to achieve significant performance improvements (Meng et al., 2024). Moreover, the simultaneous loading of two models in DPO leads to *inefficiencies* and remains a challenge, while reference-free approaches like CPO and SimPO either require additional steps or face difficulties without the SFT stage, which we refer to as the *optimization conflict* issue (See Figure 4).

Another critical challenge is *judgment noise* within preference data (Wang et al., 2024a;f), where ranking inconsistencies reduce optimization effectiveness despite improvements in open-source reward models like PairRM (Jiang et al., 2023b) and ArmoRM (Wang et al., 2024d). These models, while cost-effective, still produce inaccurate rankings that hinder direct preference optimization methods (Meng et al., 2024). Furthermore, although GPT family models help reduce noise, they remain expensive and still exhibit issues such as judgment bias (Tan et al., 2024). Finally, *reward modeling* remains an issue, as DPO relies on KL-divergence, whereas SimPO suggests that the average likelihood of response sequences provides a more effective reward representation. However, these implicit rewards behave inconsistently across varying dataset sizes, highlighting the need for more robust and scalable reward estimation methods to enhance preference optimization performance.

## 2.2 TPO: Triple Preference Optimization

Motivated by these challenges, we propose the **Triple Preference Optimization (TPO)** method, a novel preference learning approach that optimizes a pre-trained model in a single step on three preferences where $y_{\text{gold}}$, $y_w$, and $y_l$ are the preference responses generated for the same prompt $x$. To optimize a policy model with TPO, we require a dataset $D_{\text{TPO}} = \{x^i, y_{\text{gold}}^i, y_w^i, y_l^i\}_{i=1}^N$, where the preferences satisfy $y_{\text{gold}} \succ y_w \succ y_l$. This approach assumes that preference data is collected from different models, ranked by "Judge models", and used to optimize a pre-trained model. Alternatively, preference data can be derived from a Supervised Fine-Tuned (SFT) model on a task, with generated responses for the same prompt ranked by Judger models to optimize an Instruction model.

In this context, the policy model $\pi_\theta$ is not equal to the SFT model ( $\pi_\theta \neq \pi_{\text{SFT}}$), and it is optimized just in a single step for each scenario. Further details about these settings are discussed in Section 3.

### 2.2.1 Deriving the TPO objective

In this section, we detail the derivation of the TPO objective in a manner similar to the DPO derivations. We start with a basic Reinforcement Learning (RL) objective used to align a Large Language Model (LLM), parameterized by $\theta$ and represented as $\pi_\theta$, with preferences. The RL objective involves maximizing the expected reward, as defined in (Ziegler et al., 2019a) is:

$$\max_{\pi_\theta} \mathbb{E}_{x \sim \mathcal{D}, y \sim \pi_\theta(y|x)}[r_\phi(x, y)] \tag{1}$$

where $r_\phi$ represents the expected reward the model receives for a given input $x$ and output $y$. However, maximizing the reward without constraints can lead to distribution collapse in an LLM. Drawing inspiration from the Maximum Entropy Reinforcement Learning (MERL) framework (Hejna et al., 2023), we have modified the RLHF objective, as detailed in Equation 4. The MERL framework aims to maximize causal entropy alongside the expected reward. This objective is formally defined in Equation 2:

$$\max_{\pi_\theta} \mathbb{E}_{x \sim \mathcal{D}, y \sim \pi_\theta(y|x)} \left[ r_\phi(x, y) + \beta \mathcal{H}_{\pi_\theta}(y|x) \right]. \tag{2}$$

By the definition of Entropy,

$$\mathcal{H}_{\pi_\theta}(y|x) = -\frac{1}{|y|} \sum_{j=1}^{|y|} \log \pi_\theta(y_j|x, y_{<j}). \tag{3}$$

Using Equations 2 and 3 the objective becomes:

$$\max_{\pi_\theta} \mathbb{E}_{x \sim \mathcal{D}, y \sim \pi_\theta(y|x)} \left[ r_\phi(x, y) - \beta \log \pi_\theta(y|x) \right]. \tag{4}$$

Based on this, the optimal policy model induced by a reward function $r(x, y)$ could be derived as shown in Equation 5 (See Appendix A.1). It takes the following form:

$$\pi_r(y|x) = \frac{1}{Z(x)} \exp\left(\frac{1}{\beta} r(x, y)\right), \tag{5}$$

where $Z(x) = \sum_y \exp\left(\frac{1}{\beta} r(x, y)\right)$ is the new partition function. Inspired by (Rafailov et al., 2024b), we have the reward function in terms of the optimal policy that it induces; as shown below in Equation 6:

$$r(x, y) = \beta \log \pi_r(y|x) + \beta \log Z(x). \tag{6}$$

Subsequently, we can represent the ground-truth reward $r^*(x, y)$ in the form of its corresponding optimal policy $\pi^*$ that it induces.

Since the Bradley-Terry model is dependent only on the difference between the two reward functions, i.e., $p^*(y_w > y_l|x) = \sigma(r^*(x, y_w) - r^*(x, y_l))$, we can reparameterize it as follows in Equation 7:

$$p^*(y_w > y_l \mid x) = \sigma\left( \beta \log \pi^*(y_w \mid x) - \beta \log \pi^*(y_l \mid x) \right). \tag{7}$$

Similar to the reward modeling approach, we model the human preferences, which are now in terms of a parameterized policy $\pi_\theta$. Thus, we formulate maximum-likelihood objective (*preference* objective) for a dataset $D = \{x^i, y_w^i, y_l^i\}_{i=1}^N$ as shown in Equation 8:

$$\mathcal{L}_{\text{preference}}(\pi_\theta) = -\mathbb{E}_{(x,y_w,y_l) \sim \mathcal{D}} \left[ \log \sigma \Big( \beta \log \pi_\theta(y_w \mid x) - \beta \log \pi_\theta(y_l \mid x) \Big) \right]. \tag{8}$$

Looking at the Equation 8, the objective is fitting $r(x, y) = \beta \log \pi(y|x)$ as the reparameterized reward. In Section A.3 of the Appendix, we theoretically explain that fitting this reward will ultimately recover the optimal policy.

The comparison between the loss function in Equation 8 and the DPO loss function indicates that the new function is more efficient because it requires only one model during training. However, even though maximizing the objective under the MERL setting prevents distribution collapse, it trains a pessimistic model, which also limits the model from learning the preferred responses effectively. To counteract this limitation, we maximize the likelihood of the gold response. The adjustment is specified in Equation 9:

$$\mathcal{L}_{\text{reference}}(\pi_\theta) = -\mathbb{E}_{(x,y_{\text{gold}}) \sim \mathcal{D}} \left[ \log \pi_\theta \left( y_{\text{gold}} \mid x \right) \right]. \tag{9}$$

Based on Equations 8, and 9, the TPO is defined as a multi-objective (bi-objective) optimization problem supported by the Pareto Front concept (Lotov & Miettinen, 2008). The TPO loss function is thus formulated as follows:

$$\mathcal{L}_{\text{TPO}} = \mathcal{L}_{\text{preference}} + \alpha \mathcal{L}_{\text{reference}}, \tag{10}$$

where hyper-parameter ($\alpha$) is a regularization term.

Inspired by SimPO, which replaced the summation of the likelihood of response sequences with averaging, based on concepts from beam search and multiple-choice tasks, and introduced a reward margin, we propose TPO-L, a length-controlled extension of the TPO loss function. TPO-L is defined as follows:

$$\mathcal{L}_{\text{TPO-L}} = -\mathbb{E} \left[ \log \sigma \left( \frac{\beta}{|y_w|} \log \pi_\theta(y_w \mid x) - \frac{\beta}{|y_l|} \log \pi_\theta(y_l \mid x) - \gamma \right) - \alpha \log \pi_\theta(y_{\text{gold}} \mid x) \right], \tag{11}$$

where, $\gamma$ represents the reward margin. In Section 4.3, we will analyze the reward margin's impact, showing it provides a superior reward representation than SimPO. We will also demonstrate that while summation in the loss function promotes longer responses, a well-fitted policy model generates effective responses without depending on summation.

## 3 Experimental Setup

**Models and training settings.** To evaluate TPO and existing preference optimization methods, we follow the SimPO setup with minor adjustments, focusing on the Mistral Jiang et al. (2023a) and LLaMA Touvron et al. (2023) models. To simplify the main text, we omit Mistral results but include them in Appendix D. Our evaluation consists of two distinct setups: **Base** and **Instruct**.

**Base setting.** For this setting, we used the UltraFeedback (Cui et al., 2023) dataset, containing 60,000 data points, each with four responses scored by GPT-4 across four criteria, resulting in an overall score. As TPO requires three preferences ($y_{\text{gold}} \succ y_w \succ y_l$), we processed the dataset to ensure a 0.5 score difference between $y_{\text{gold}}$, $y_w$, and $y_l$ (See Figure 9 in Appendix). This resulted in a final dataset of 40,000 data points. To compare preference optimization methods fairly, we fine-tuned a pre-trained model on the gold responses and used the preferred and rejected responses for the current preference optimization methods in two steps. For TPO, we utilized all data in one optimization step. Moreover, we evaluated preference optimization methods on subsets of 5,000, 10,000, and 20,000 points randomly selected from the processed dataset. The Experiment section 4.1 details the models' performance.

Table 1: Evaluation details for Arena-Hard, MT-Bench, and MixEval-Hard as the instruction following benchmarks and MMLU-Pro, MMLU, and GSM8K as the reasoning benchmarks.

| | # Exs. | Baseline | Evaluation | Scoring Type | Metric |
|---|---|---|---|---|---|
| **Arena-Hard** | 500 | Answer of GPT-4-0314 | Judge by GPT-4o | Pairwise comparison | Win rate |
| **MT-Bench** | 80 | - | Judge by GPT-4 | Single-answer grading | Rate of 1-10 |
| **MixEval-Hard** | 1,000 | Ground truth | Evaluate by GPT-3.5-turbo | Systematic | Accuracy |
| **MMLU-Pro** | 12,032 | Ground truth | CoT | Systematic | Accuracy |
| **MMLU** | 15,908 | Ground truth | 5-shots | Systematic | Accuracy |
| **GSM8K** | 1,319 | Ground truth | 5-shots | Systematic | Accuracy |

**Instruction setting.** We used instruction-tuned models as the backbone for preference optimization to assess the impact of replacing them with SFT models. The experiment employed the UltraFeedback-ArmoRM dataset, comprising 60,000 samples with five responses generated by the LLaMA-3-8B-SFT, fine-tuned on UltraChat (Ding et al., 2023b). Responses were scored using the ArmoRM reward model, with the highest and lowest-scoring responses designated as preferred ($y_w$) and rejected ($y_l$), respectively. For TPO, the highest-scoring response was considered gold ($y_{gold}$), the second-highest as preferred ($y_w$), and the lowest as rejected ($y_l$). The best checkpoint from SimPO was used for comparison.

**Evaluation benchmarks.** We categorized the evaluation benchmarks into reasoning and instruction-following tasks. Reasoning benchmarks included MMLU (Hendrycks et al., 2021), MMLU-pro (Wang et al., 2024e), and GSM8K (Cobbe et al., 2021), which assess a model's understanding across diverse subjects, with MMLU-pro featuring more complex tasks and GSM8K focusing on math problem-solving requiring step-by-step reasoning. Instruction-following benchmarks consisted of Arena-Hard (Li et al., 2024), MT-Bench (Zheng et al., 2023b), and MixEval-Hard (Ni et al., 2024), which evaluate conversational versatility across a range of queries. MT-Bench includes 80 questions across 8 categories, Arena-Hard expands on it with 500 technical problem-solving queries, and MixEval-Hard tests challenging queries against known answers using GPT-3.5-turbo for verification (details in Table 1). Additionally, we reported scores for ARC (Clark et al., 2018), HellaSwag (Zellers et al., 2019), Winogrande (Levesque et al., 2012), and TruthfulQA (Lin et al., 2022) in a standard setup, with further details provided in Appendix E.

**Baselines.** We compare TPO with several preference optimization methods, including SLiC-HF (Zhao et al., 2023), which uses ranking losses, and IPO (Azar et al., 2023), which avoids DPO's pointwise reward assumption with a theoretically grounded approach. SimPO (Meng et al., 2024), a reward margin-aware variant of DPO, leverages average sequence likelihood instead of summation. CPO (Xu et al., 2024) integrates training with an SFT objective, while KTO (Ethayarajh et al., 2024) learns from non-paired preference data. ORPO (Hong et al., 2024a) eliminates the need for a reference model by introducing an odds ratio term, directly comparing preferred and rejected responses while jointly training with SFT. SimPO findings suggest ORPO performs better when fine-tuned from an SFT checkpoint. We fine-tuned hyperparameters for all baselines. Additional details are available in Appendix C.

# 4 Experimental Results

This section presents the main results across benchmarks, comparing TPO with existing preference optimization methods at different dataset sizes. We analyze reward modeling in TPO and compare it with SimPO, address the verbosity issue by showing TPO and TPO-L generate shorter responses, and evaluate TPO's performance on noisy data compared to DPO. Additionally, we assess the effect of strong SFT checkpoints, optimized on larger datasets, on DPO's performance. Further details are provided in Appendix E.

## 4.1 Main Results and Ablations

**TPO and TPO-L significantly outperform existing preference optimization methods.** The results in Tables 2 and 3 demonstrate that TPO and TPO-L significantly outperform existing preference optimization methods across reasoning and instruction-following benchmarks in both Base and Instruct settings, regardless of training dataset size. These

Table 2: Results for Arena-Hard, MT-Bench, MixEval-Hard, MMLU-Pro, MMLU, and GSM8K under the Base setting across four training sizes. LLaMA3-8B-Base models are trained on gold responses and fine-tuned with preference optimization on chosen and rejected data, highlighting TPO and TPO-L improvements over DPO. We quantified the improvement over DPO alongside the TPO and TPO-L scores.

| Method | UltraFeedback (5k) | | | | | | Ultrafeedback (10k) | | | | | |
| | Reasoning | | | Instruction Following | | | Reasoning | | | Instruction Following | | |
| | GSM8K | MMLU-Pro | MMLU | MT-Bench | Arena-Hard | MixEval-Hard | GSM8K | MMLU-Pro | MMLU | MT-Bench | Arena-Hard | MixEval-Hard |
|---|---|---|---|---|---|---|---|---|---|---|---|---|
| SFT | 28.5 | 25.7 | 59.0 | 4.5 | 0 | 24.6 | 27.7 | 25.5 | 59.1 | 4.9 | 0 | 24.4 |
| DPO | 32.9 | 27.1 | 59.2 | 5.5 | < 0.5 | 26.9 | 36.7 | 28.9 | 59.4 | 6.3 | 6.1 | 25.9 |
| CPO | 31.9 | 26.5 | 58.9 | 5.4 | < 0.5 | 27.4 | 32.8 | 26.9 | 59.3 | 5.5 | < 0.5 | 23.9 |
| IPO | 34.4 | 28.0 | 59.4 | 5.6 | < 0.5 | 27.1 | 35.4 | 28.3 | 59.2 | 6.4 | 3.9 | 26.7 |
| ORPO | 34.0 | 27.5 | 59.3 | 5.0 | < 0.5 | 28.3 | 36.2 | 28.1 | 59.7 | 5.0 | < 0.5 | 27.4 |
| KTO | 32.7 | 26.8 | 59.2 | 5.5 | < 0.5 | 26.5 | 36.7 | 28.8 | 59.4 | 6.4 | 4.4 | 27.2 |
| SIMPO | 32.7 | 27.4 | 59.2 | 5.6 | < 0.5 | 27.4 | 35.1 | 27.5 | 59.3 | 5.7 | 4.4 | 24.8 |
| SLIC-HF | 32.5 | 26.7 | 59.0 | 5.4 | < 0.5 | 26.5 | 32.7 | 26.7 | 59.3 | 5.5 | < 0.5 | 25.5 |
| TPO | 51.9 (+19.0) | 37.5 (+10.4) | 65.3 (+6.1) | 6.2 (+0.7) | 4.4 (+3.9) | 35.0 (+8.1) | 52.2 (+15.5) | 37.8 (+8.9) | 65.3 (+5.9) | 6.7 (+0.4) | 6.7 (+0.6) | 31.2 (+5.3) |
| TPO-L | 52.1 (+19.2) | 36.3 (+9.2) | 65.3 (+6.1) | 6.3 (+0.8) | 2.0 (+1.5) | 31.8 (+4.9) | 51.9 (+15.2) | 38.1 (+9.2) | 65.4 (+6.0) | 6.9 (+0.6) | 7.3 (+1.2) | 33.7 (+7.8) |

| Method | UltraFeedback (20k) | | | | | | Ultrafeedback (40k) | | | | | |
| | Reasoning | | | Instruction Following | | | Reasoning | | | Instruction Following | | |
| | GSM8K | MMLU-Pro | MMLU | MT-Bench | Arena-Hard | MixEval-Hard | GSM8K | MMLU-Pro | MMLU | MT-Bench | Arena-Hard | MixEval-Hard |
|---|---|---|---|---|---|---|---|---|---|---|---|---|
| SFT | 20.3 | 30.4 | 62.5 | 5.5 | 0 | 28.8 | 39.2 | 29.5 | 62.2 | 5.5 | 1.7 | 25.5 |
| DPO | 48.0 | 35.7 | 62.8 | 6.5 | 2 | 30.1 | 45.1 | 33.5 | 61.8 | 6.6 | 5.3 | 25.4 |
| CPO | 45.0 | 32.6 | 62.9 | 6.3 | 7.1 | 30.5 | 42.9 | 30.4 | 62.2 | 6.2 | 9.3 | 24.4 |
| IPO | 45.5 | 35.0 | 62.8 | 5.9 | < 0.5 | 26.2 | 48.2 | 34.2 | 62.1 | 6.7 | 4.2 | 27.1 |
| ORPO | 37.1 | 33.1 | 63.1 | 5.6 | < 0.5 | 28.9 | 40.9 | 32.1 | 62.4 | 5.9 | 3.2 | 27.7 |
| KTO | 48.7 | 35.8 | 62.7 | 6.6 | 1.7 | 29.0 | 46.0 | 33.4 | 61.9 | 6.6 | 5.2 | 24.0 |
| SIMPO | 45.0 | 34.0 | 63.0 | 6.4 | 5.1 | 28.1 | 45.1 | 31.6 | 61.9 | 6.2 | 6.6 | 24.3 |
| SLIC-HF | 44.1 | 32.5 | 62.9 | 6.3 | 7.3 | 30.1 | 42.8 | 30.5 | 62.3 | 6.4 | 3 | 25.8 |
| TPO | 53.0 (+5.0) | 37.5 (+1.8) | 65.3 (+2.5) | 6.6 (+0.1) | 5.3 (+3.3) | 32.4 (+2.3) | 51.2 (+6.1) | 37.4 (+3.9) | 64.8 (+3.0) | 6.9 (+0.3) | 6.9 (+1.6) | 32.9 (+7.5) |
| TPO-L | 52.7 (+4.7) | 38.1 (+2.4) | 65.3 (+2.5) | 6.8 (+0.3) | 6.5 (+4.5) | 32.4 (+2.3) | 52.4 (+7.3) | 40.4 (+6.9) | 65.1 (+3.3) | 7.3 (+0.7) | 12.3 (+7.0) | 37.6 (+12.2) |

findings highlight that TPO not only enhances the instruction-following capabilities of models but also significantly improves their reasoning performance.

TPO and TPO-L excel in low-data scenarios, making them ideal for tasks with limited dataset availability. Notably, with only 5,000 samples, other methods perform poorly on Arena-Hard, while TPO and TPO-L achieve 4.4% and 2%, respectively. TPO-L outperforms TPO in the Base setting, whereas TPO delivers better results in the Instruct setting, highlighting their adaptability and robustness across various benchmarks and training setups.

**TPO-L is more stable and generalizable than other methods for downstream tasks.** We used the DAA metric to evaluate the generalizability of competitor methods across five downstream tasks: GSM8K, MMLU, HellaSwag, Winogrande, and TruthfulQA. The results shown in Figure 2 indicate that while existing methods provide slight improvements over the SFT checkpoint, their effectiveness is significantly influenced by dataset size. In the instruction setting, methods such as DPO and IPO

Table 3: Results for Are-Hard, MT-Bench, MixEval-Hard, MMLU-Pro, MMLU, and GSM8K under the Instruct setting, using off-the-shelf models as the SFT model. Front of the TPO and TPO-L scores, we identify the improvement compared with DPO.

| Method | LLaMA-3B-Instruct | | | | | |
| | Reasoning | | | Instruction Following | | |
| | GSM8K | MMLU-Pro | MMLU | MT-Bench | Arena-Hard | MixEval-Hard |
|---|---|---|---|---|---|---|
| SFT[1] | 68.7 | - | 67.1 | 8.1 | - | - |
| DPO[1] | 54.5 | 43.7 | 67.3 | 7.7 | 40.9 | 41.6 |
| CPO[1] | 67.8 | 42.6 | 66.9 | 7.9 | 35.6 | 41.3 |
| IPO[1] | 56.3 | 42.8 | 67.3 | 7.9 | 40.1 | 44.0 |
| ORPO[1] | 63.4 | 42.7 | 66.8 | 7.9 | 34.8 | 36.4 |
| KTO[1] | 66.4 | 42.1 | 67.2 | 8.1 | 34.7 | 44.1 |
| SIMPO[1] | 55.6 | 40.5 | 66.5 | 7.7 | 35.1 | 45.0 |
| SLIC-HF[1] | 68.2 | 40.2 | 66.9 | 7.7 | 28.6 | 40.0 |
| $TPO_{y2}$ | 77.2 | 44.4 | 65.9 | 8.2 | 42.0 | 40.4 |
| $TPO_{y3}$ | 77.8 | 43.8 | 65.5 | 8.2 | 42.4 | 41.9 |
| $TPO_{y4}$ | 78.2 | 43.4 | 65.7 | 8.1 | 38.9 | 38.6 |
| $TPO\text{-}L_{y2}$ | 77.3 | 43.7 | 65.7 | 8.2 | 39.4 | 39.4 |

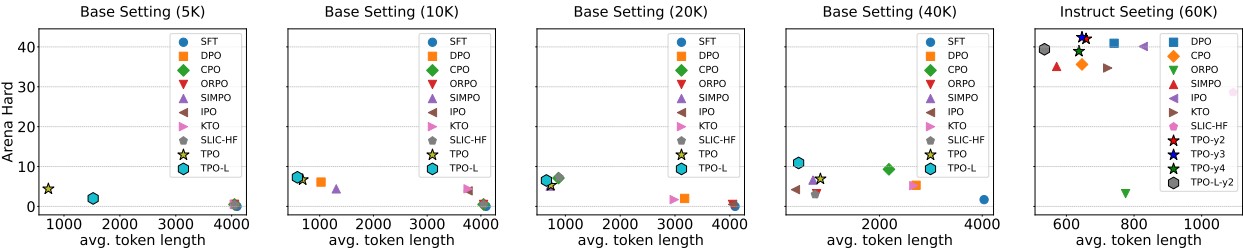

Figure 3: Comparison of Arena-Hard scores and average token length of responses across 500 prompts for different methods.

failed to match the performance of the SFT checkpoint, highlighting their limited impact on downstream tasks. Conversely, TPO and TPO-L achieved substantial improvements over other methods. However, TPO displayed performance instability across various dataset sizes, while TPO-L consistently provided stable enhancements, attaining the highest DAA values in most of the Base settings. This emphasizes TPO-L's superior stability and generalizability compared to other methods.

**TPO demonstrates higher effectiveness in a preference dataset.** Recent frameworks have improved the efficiency of generating multiple high-quality responses (Kwon et al., 2023). However, the current two-preference methods may lose valuable data when responses have similar quality. TPO addresses this issue by utilizing three preferences, which use more data and improve optimization. Consider a dataset $D = \{x, y_1, y_2, y_3, y_4, y_5\}$, where $y_1 \succ y_2 \succ y_3 \succ y_4 \succ y_5$. We fine-tuned models under three combinations of preferences: $\{x, y_1, y_2, y_5\}$, where $y_1 = y_{\text{gold}}, y_2 = y_w, y_5 = y_l$; $\{x, y_1, y_3, y_5\}$, where $y_3 = y_w$; and $\{x, y_1, y_4, y_5\}$, where $y_4 = y_w$. The results in Table 3 show that TPO in the second combination achieves the best performance, likely because of its resilience to judgment noise in the dataset. This demonstrates TPO's capability to exceed the current two-preference methods.

## 4.2 Verbosity Problem

Recent studies (Park et al., 2024; Meng et al., 2024) identify verbosity as a key challenge in alignment algorithms, where models produce overly long responses during post-training without quality gains. SimPO addresses this by replacing sequence summation in the DPO loss with average likelihood, reducing verbosity. In this section, we analyze how TPO and TPO-L mitigate this issue and assess their effectiveness.

**The sequence summation in TPO does not lead to verbosity.** To analyze verbosity, we used the Length Control setup from Arena-Hard. As shown in Figure 3, TPO consistently ranks among the top three methods for average token length across all settings. Notably, even in the 5,000 Base setting, TPO generates shorter responses while achieving the best performance. Further analysis of token lengths revealed that gold responses are not necessarily shorter than preferred or rejected ones, suggesting that TPO mitigates verbosity by enhancing the model's understanding rather than just shortening responses. More details are available in Appendix G. Overall, TPO with the summation term does not exhibit verbosity issues, challenging the findings reported in the SimPO paper.

**TPO-L improves the length control ability of the TPO.** Figure 3 shows that TPO-L enhances TPO's length control ability across the Base settings, except for the 5,000 setting. A comparison of response length and Arena-Hard performance between TPO-L and SimPO reveals that incorporating the average likelihood of the response sequence has a stronger effect on the TPO loss function compared to the DPO loss function.

## 4.3 Analysis of TPO

This section explores optimization conflict in CPO compared to TPO and analyzes reward modeling in TPO and TPO-L. We examine the reward distribution of TPO against SimPO and SFT checkpoints across different training sizes and

---

[1]We evaluated the latest checkpoints reported in the SimPO repository across benchmarks and used GSM8K and MMLU values from the SimPO paper.

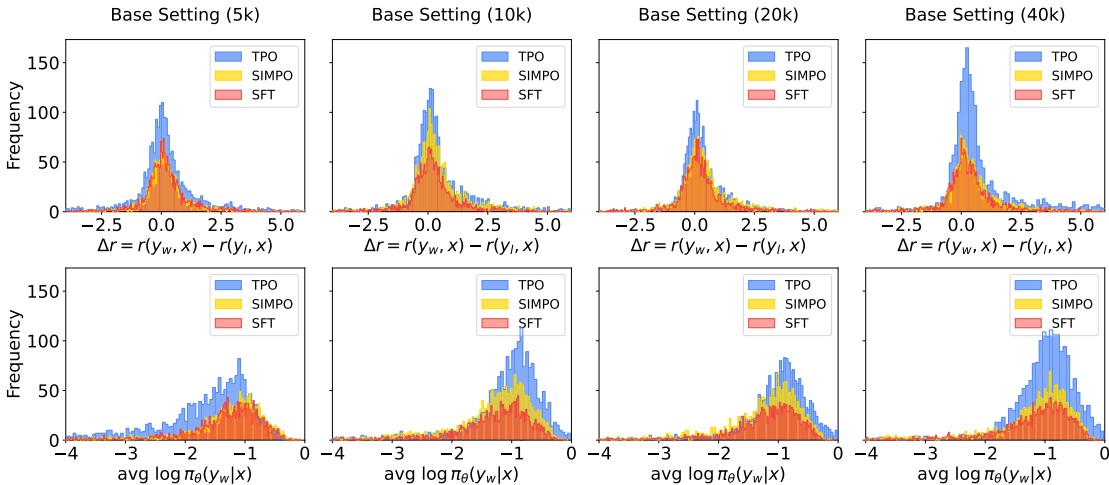

Figure 5: Reward modeling analysis on the UltraFeedback test set. **Top:** Comparison of reward distributions for SFT, SimPO, and TPO across different data sizes. **Bottom:** Evaluation of $\log \pi_\theta(y|x)$ as an implicit reward across varying data sizes.

evaluate the impact of reward margin $\gamma$ adjustments in TPO-L on reward accuracy and model performance. Further analyses are provided in Appendix B.

**TPO resolves the optimization conflict issue in CPO.** If $y_{\text{gold}} = y_w$ and $\alpha = 1$, the TPO loss function reduces to the CPO loss function. However, TPO consistently outperforms CPO across benchmarks and settings. We analyze why TPO achieves better performance than CPO. Theoretical analysis of the CPO loss function shows that it incorporates a Behavioral Cloning (BC) objective into preference optimization to maintain the policy model within the distribution of preferred responses. It was hypothesized that adding a BC objective for preferred data would improve the increasing likelihood of preferred responses during optimization. However, as observed in prior work (Rafailov et al., 2024a), and supported by our

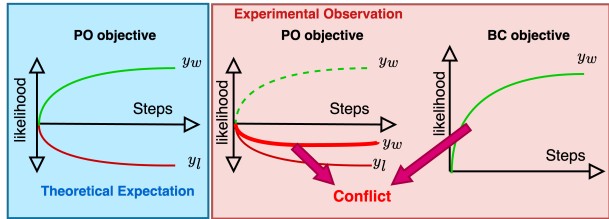

Figure 4: Difference between the theoretical expectation and the experimental observation of the Preference Optimization (PO) objective part in the CPO loss function during optimization.

findings, a conflict arises where the likelihood of $y_w$ decreases during the optimization process in CPO, contradicting the intended objective. As shown in Figure 4, although the BC objective is designed to increase the likelihood of $y_w$ in preference optimization, the opposite occurs, with $y_w$'s likelihood diminishing instead. We refer to this phenomenon as *optimization conflict*.

**TPO enhances the reward distribution as the training data size increases.** To compare the reward distribution of TPO and SimPO, we calculate avg. $\log \pi_\theta(y_w|x)$ and $\Delta r = r(y_w, x) - r(y_l, x)$, where $y_w$ and $y_l$ denote the preferred and rejected responses in the UltraFeedback test set. As shown in Figure 5, TPO demonstrates a consistent increase in the average likelihood of $y_w$ with larger training data sizes, whereas SimPO shows only a slight improvement. Additionally, the comparison of $\Delta r$ demonstrates that TPO maintains a broader reward distribution across all dataset sizes compared to SimPO. Overall, TPO effectively enhances reward distribution as the training set size grows.

**Increasing the reward margin in TPO-L enhances the reward distribution.** To examine the impact of the reward margin in TPO-L, we analyzed the 40,000 Base setting by calculating reward accuracy and visualizing distributions on the UltraFeedback test set. Figure 6 shows that increasing the reward margin extends the $\Delta r$ distribution. While the average likelihood of $y_w$ initially decreases up to a margin of 5.4, further increases encourage the model to enhance the average likelihood of $y_w$.

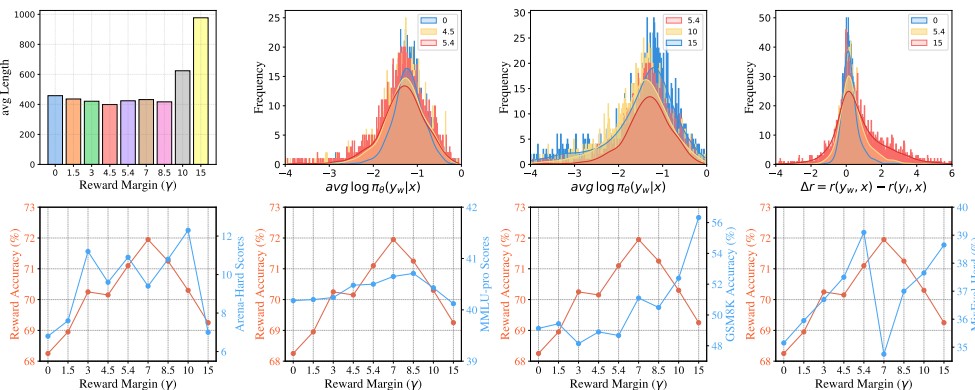

Figure 6: Study of the reward margin $\gamma$. **Top:** Reward distribution under different $\gamma$ values on UltraFeedback test set. **Bottom:** Reward accuracy on the UltraFeedback test set and accuracy on Arena-Hard, MMLU-Pro, GSM8K, and MixEval-Hard under different $\gamma$ values.

**Increasing the reward margin vs different benchmarks.** The results in Figure 6 show that increasing the reward margin significantly enhances GSM8K accuracy but leads to a substantial performance drop on Arena-Hard at higher margins. Additionally, a higher reward margin encourages longer responses, indicating that the average likelihood approach alone cannot fully control verbosity. Minimal changes were observed in MMLU-Pro performance across different reward margins, suggesting a limited impact. MixEval-Hard results indicate optimal performance at a reward margin of 5.4, with a sharp decline beyond this point.

## 4.4 Robustness of TPO and DPO

This section investigates the robustness of TPO and DPO in various scenarios. We analyze the impact of judgment noise in preference datasets, consider real-world conditions where two out of three responses receive identical scores, and evaluate DPO's performance when initialized from a more robust SFT checkpoint.

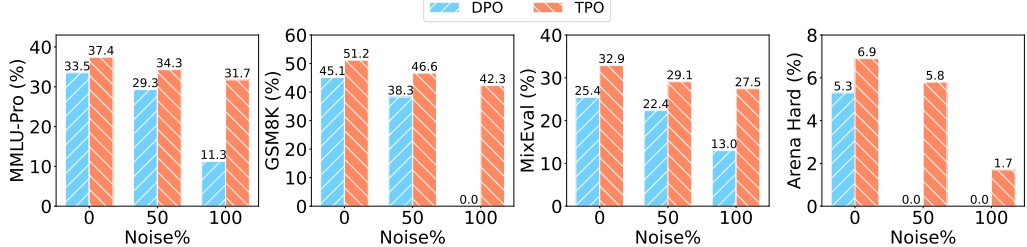

Figure 7: Robustness comparison of TPO and DPO under different percentages of judgment noise.

**TPO displays stronger robustness to judgment noise than DPO.** To evaluate TPO and DPO under noisy conditions, we use the 40,000 Base setting, assuming that judgment noise affects the preference dataset while the gold dataset remains clean. The SFT checkpoint for DPO is fine-tuned on the clean gold dataset. Noise is defined as incorrect decisions between preferred and rejected responses, with 50% and 100% noise levels introduced into the training set.

Figure 7 shows that as noise increases, DPO performance collapses, with scores on GSM8K and Arena-Hard dropping to zero under 100% noise. In contrast, TPO maintains acceptable performance despite some decline across benchmarks. This stability is attributed to the $\log \pi_\theta(y_{\text{gold}}|x)$ term in TPO, which leverages the gold response to mitigate the impact of noisy preference data. These findings highlight the importance of the BC objective on the gold response in maintaining model performance under noisy conditions.

**TPO performs reliably across different hypotheses.** TPO assumes the dataset satisfies $y_{\text{gold}} \succ y_w \succ y_l$, but constructing such data can be challenging when two preferences have similar scores. To evaluate TPO under these conditions, we created a 10,000-example dataset from UltraFeedback, ensuring $S(y_{\text{gold}}) = S(y_w)$, with a 0.5-point difference from the rejected response. Results in Table 4 show that TPO performs similarly in this setup to its original assumption. This finding suggests that even when $y_{\text{gold}}$ and $y_w$ share the same score, using distinct responses for $y_{\text{gold}}$ and $y_w$ yields better performance compared to the case where $y_{\text{gold}} = y_w$, which is similar to the CPO loss function.

Table 4: Comparison of TPO assuming gold and preferred responses differ but share the same quality score.

| Method | Quality Comparison | GSM8k | MMLU-Pro | MMLU | MT-Bench | Arena-Hard (Avg. # TL) | MixEval-Hard |
|--------|--------------------|-------|----------|------|----------|------------------------|--------------|
| TPO | $y_{gold} \succ y_w \succ y_l$ | **52.2** | 37.8 | **65.3** | **6.7** | 6.7 (683) | 31.2 |
| TPO | $y_{gold} \simeq y_w \succ y_l$ | 51.5 | **37.8** | 65.3 | 6.4 | **8.2 (642)** | **32.0** |

**TPO is more effective than DPO while using less data.** Recent studies (Tunstall et al., 2023; Saeidi et al., 2025) emphasize the importance of the SFT checkpoint in DPO, as it can be fine-tuned with additional data to enhance performance. In this study, we evaluated DPO's performance using different, stronger SFT checkpoints fine-tuned on 20,000 and 40,000 data points within the UltraFeedback 10,000 Base setting. Results in Table 5 show that while DPO fine-tuned on stronger SFT datasets improved performance, especially on Arena-Hard, its reasoning benchmark scores declined with the 40,000 dataset compared to the 40,000 Base setting. Moreover, despite using twice as much data as TPO and TPO-L, DPO only surpassed them on Arena-Hard, demonstrating that TPO achieves comparable performance with significantly less data, highlighting its efficiency.

Table 5: Comparison of TPO and TPO-L with DPO, which uses more robust SFT checkpoints.

| Method | SFT Data | Preference Data | GSM8k | MMLU-Pro | MMLU | MT-Bench | Arena-Hard | MixEval-Hard |
|--------|----------|-----------------|-------|----------|------|----------|------------|--------------|
| DPO | 10K ($y_{gold}$) | 10K ($y_w, y_l$) | 36.7 | 28.8 | 59.4 | 6.3 | 6.1 | 25.9 |
| DPO | 20K ($y_{gold}$) | 10K ($y_w, y_l$) | 49.8 | 35.8 | 63.0 | 6.6 | 7.1 | 28.5 |
| DPO | 40K ($y_{gold}$) | 10K ($y_w, y_l$) | 46.8 | 32.3 | 62.3 | 6.6 | **7.8** | 25.2 |
| TPO | none | 10K ($y_{gold}, y_w, y_l$) | **52.2** | 37.8 | 65.3 | 6.7 | 6.7 | 31.2 |
| TPO-L | none | 10K ($y_{gold}, y_w, y_l$) | 51.9 | **38.0** | **65.4** | **6.9** | 7.3 | **33.7** |

# 5 Related Works

**Reinforcement learning from human feedback.** RLHF is used to align large language models with human preferences and values (Christiano et al., 2017; Ziegler et al., 2019b; Ouyang et al., 2022a; Bai et al., 2022). A classical RLHF setting involves three stages: supervised fine-tuning stage (Zhou et al., 2023a; Taori et al., 2023; Geng et al., 2023; Conover et al., 2023; Köpf et al., 2023; Ding et al., 2023a; Wang et al., 2024b; Chen et al., 2024a; Xia et al., 2024), reward modeling stage (Gao et al., 2023; Luo et al., 2023; Chen et al., 2024b; Lightman et al., 2023; Havrilla et al., 2024b; Lambert et al., 2024), and policy optimization stage (Schulman et al., 2017; Anthony et al., 2017). During the final stage, Proximal Policy Optimization (PPO) (Schulman et al., 2017) is more commonly used. This approach involves optimizing for maximum reward by interacting with a reward model trained using the Bradley-Terry objective. The application of the RLHF framework transcends various applications, such as ensuring safety (Dai et al., 2023), enhancing helpfulness (Tian et al., 2024; Wang et al., 2024c), mitigating toxicity (Amini et al., 2024; Korbak et al., 2023; Zheng et al., 2023a), searching and navigating the web (Nakano et al., 2021), and improving model reasoning abilities (Havrilla et al., 2024a). While Reinforcement Learning with Human Feedback (RLHF) enhances model performance, it encounters challenges such as the collection of preference data (Casper et al., 2023), unstable training processes, and the generation of biased or overly verbose responses (Dubois et al., 2024; Singhal et al., 2023; Wang et al., 2023).

**Offline vs. Iterative preference optimization.** Owing to the challenges and complexity of online preference optimization techniques(Zheng et al., 2023c; Santacroce et al., 2023), recent works have proposed simple and efficient offline algorithms. One among them is Direct Preference Optimization (DPO)(Rafailov et al., 2023). It makes use of an

implicit reward, which is in terms of the parameterized policy and a reference policy. Due to the absence of an explicit reward model, DPO is limited in its ability to sample preferences from the optimal policy. Addressing this, works like (Zhao et al., 2023; Liu et al., 2024b) explore augmenting preference data using a trained or refined SFT policy. Iterative training methods, which continuously update the reference model with the most recent policy model or generate new preference pairs at each iteration, have been explored by works (Dong et al., 2024; Kim et al., 2024; Rosset et al., 2024; Xiong et al., 2024; Yuan et al., 2024; Xie et al., 2024; Cen et al., 2024). In this work, we experiment in an offline setting, without any iterative training.

**Preference optimization objectives.** Besides DPO, various preference optimization algorithms have been proposed. A variety of preference optimization objectives have been proposed besides DPO (Dong et al., 2023; Liu et al., 2024a; Song et al., 2024; Yuan et al., 2023). However, this family of methods either relies on explicit rewards during preference optimization or demands a fine-tuned reference model. In contrast, TPO avoids this by using three preferences. Similar to TPO, simpler objectives that do not rely on a reference model have been proposed (Hong et al., 2024a; Xu et al., 2023). In this work, we compare TPO to a series of offline algorithms, including RRHF Yuan et al. (2023), SLiC-HF Zhao et al. (2023), DPO Rafailov et al. (2023), IPO Azar et al. (2023), CPO Xu et al. (2024), KTO Ethayarajh et al. (2024), ORPO Hong et al. (2024a), and SimPO Meng et al. (2024), and find that TPO can outperform them in both efficiency and performance. Similar to TPO, SimPO does not impose KL regularization, however, the robustness of its effective learning is dependent on various factors like preference datasets with diverse domain coverage. Whereas TPO is not constrained by this.

# 6 Conclusion

In this study, we introduced the Triple Preference Optimization (TPO) method, a one-step preference optimization approach that enhances both the instruction-following and reasoning abilities of the policy model. We developed TPO-L, a length-controlled variant, by defining a reward margin and replacing summation with the average response sequence. Our results show that TPO and TPO-L outperform existing methods across benchmarks, effectively address the verbosity problem, and are less sensitive to dataset size, particularly in the case of TPO-L. Additionally, TPO demonstrates greater robustness to noisy data and performs better on less data than DPO.

## Limitations and Future works

**Exploring the Quality Margin Between Responses.** The main hypothesis of TPO focuses on maintaining a margin difference between gold and preferred responses. Although we evaluated TPO in scenarios where the gold and preferred responses share the same quality score, further exploration of the margin between gold and preferred responses, as well as between preferred and rejected responses, is still needed. Our analysis of the UltraFeedback dataset indicates that increasing the margin between gold and preferred responses significantly reduces the number of qualifying samples, leaving only 10,000 that meet the conditions. Therefore, a more in-depth analysis of this area is necessary.

**Exploration on Reward Margin.** Although TPO-L demonstrated impressive performance across various settings, it requires considerable effort to determine the optimal value for the reward margin $\gamma$. This limitation restricts the applicability of TPO-L and similar methods, such as SimPO, across different tasks. Developing dynamic approaches to adjust the reward margin automatically based on the specific task could be an interesting direction for future work.

**Safety and Honesty.** As discussed in Section 2.2, the TPO loss function incorporates two objective functions. Recent studies (Dai et al., 2023; Zhou et al., 2023b) have explored multi-objective preference optimization methods, which require preparing multiple preference pairs. Safety is an important domain where multiple objectives, such as safety and helpfulness, are equally critical. Exploring the impact of TPO on such multi-objective tasks could provide valuable insights. A key advantage of TPO compared to other methods is that data collection for TPO is significantly easier than for multi-objective methods like DPO.

**In-Depth Exploration of Reasoning.** As shown in Tables 2 and 12, TPO and TPO-L achieved strong results on reasoning benchmarks. However, to further demonstrate TPO's effectiveness on reasoning tasks, we encourage researchers to focus on reasoning-specific datasets such as UnSeenTimeQA Uddin et al. (2025) and explore preference

optimization designed for reasoning tasks. Developing and evaluating reasoning-based preference datasets would provide deeper insights into TPO's capabilities.

**Online Learning.** Recent study (Park et al., 2024) suggests that online versions of DPO address the overfitting challenges associated with offline versions, often achieving better performance. Exploring this area could be particularly valuable, as superior results on datasets like UltraFeedback-ArmoRM, which closely align with the concept of online policy learning, support the feasibility of online learning approaches.

## Ethical Considerations

The authors state that in this work, AI assistants, specifically Grammarly and ChatGPT, were utilized to correct grammatical errors and restructure sentences.

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

# Appendix

## A Derivation

### A.1 Deriving the optimal policy under the Preference Objective

In this section, we derive the optimal policy achieved by optimizing the objective in Equation 4 from Section 2.2.1. For a given prompt $x$, the objective can be analogously written as follows:

$$\max_{\pi} \mathbb{E}_{y \sim \pi(y|x)} \left[ r(x, y) - \beta \log \pi(y|x) \right] s.t. \sum_{y} \pi(y|x) = 1.$$

Next, we form a Lagrangian for the above objective with $\lambda$ being the Lagrangian multiplier.

$$\mathcal{L} = \sum_{y} \pi(y|x) r(x, y) - \beta \left[ \sum_{y} \pi(y|x) \log \pi(y|x) \right] - \lambda \left[ 1 - \sum_{y} \pi(y|x) \right].$$

Differentiating $\mathcal{L}$ with respect to $\pi(y|x)$ results in,

$$\frac{\partial \mathcal{L}}{\partial_{\pi(y|x)}} = r(x, y) - \beta \left[ \log \pi(y|x) + 1 \right] - \lambda.$$

To obtain the optimal policy, we can set the above equation to zero and solve for $\pi(y|x)$.

$$r(x, y) - \beta \left[ \log \pi(y|x) + 1 \right] - \lambda = 0,$$

$$\log \pi(y|x) = \frac{1}{\beta} r(x, y) - \frac{\lambda}{\beta} - 1,$$

$$\pi(y|x) = \exp\left(\frac{1}{\beta} r(x, y)\right) . \exp\left(\frac{-\lambda}{\beta} - 1\right).$$

Since $\sum_{y} \pi(y|x) = 1$, the second exponent is a partition function that does normalization as shown below:

$$\left[ \sum_{y} \exp\left(\frac{1}{\beta} r(x, y)\right) \right] . \exp\left(\frac{-\lambda}{\beta} - 1\right) = 1,$$

$$\exp\left(\frac{-\lambda}{\beta} - 1\right) = \left[ \sum_{y} \exp\left(\frac{1}{\beta} r(x, y)\right) \right]^{-1}.$$

Hence, the partition function $Z(x) = \sum_{y} \exp\left(\frac{1}{\beta} r(x, y)\right)$ and the optimal policy $\pi_r(y|x)$ induced by reward function $r(x, y)$ is therefore given by,

$$\pi_r(y|x) = \frac{1}{Z(x)} \exp\left(\frac{1}{\beta} r(x, y)\right). \tag{1}$$

Now, we can express the reward function in terms of an optimal policy $\pi_r$ by performing some algebraic transformations on Equation 1 as shown below,

$$\pi_r(y|x) . Z(x) = \exp\left(\frac{1}{\beta} r(x, y)\right).$$

Taking logarithm and multiplying by $\beta$ on both sides,

$$r(x,y) = \beta \log \pi_r(y|x) + \beta \log Z(x). \tag{2}$$

## A.2 Deriving the Gradient of the TPO Objective

In this section, we derive the gradient of the TPO objective:

$$\nabla_\theta \mathcal{L}_{\text{TPO}} = -\nabla_\theta \mathbb{E}_{(x,y_{ref},y_w,y_l)\sim\mathcal{D}} \left[ \alpha \log \pi_\theta(y_{ref}|x) + \log \sigma(\beta \log \pi_\theta(y_w|x) - \beta \log \pi_\theta(y_l|x)) \right]. \tag{1}$$

We can rewrite the RHS of the Equation 1 as

$$\nabla_\theta \mathcal{L}_{\text{TPO}} = -\mathbb{E}_{(x,y_{ref},y_w,y_l)\sim\mathcal{D}} \left[ \underbrace{\alpha \nabla_\theta \log \pi_\theta(y_{ref}|x)}_{(a)} + \underbrace{\nabla_\theta \log \sigma(\beta \log \pi_\theta(y_w|x) - \beta \log \pi_\theta(y_l|x))}_{(b)} \right]. \tag{2}$$

In equation 2, the part (b) can be rewritten with

$$u = \beta \log \pi_\theta(y_w|x) - \beta \log \pi_\theta(y_l|x),$$

$$\nabla_\theta \log \sigma(u) = \frac{1}{\sigma(u)} \nabla_\theta \sigma(u),$$

$$\nabla_\theta \log \sigma(u) = \frac{\sigma'(u)}{\sigma(u)} \nabla_\theta(u).$$

Using the properties of sigmoid function function $\sigma'(u) = \sigma(u)(1 - \sigma(u))$ and $\sigma(-u) = 1 - \sigma(u)$,

$$\nabla_\theta \log \sigma(u) = \frac{\sigma(u)(1 - \sigma(u))}{\sigma(u)} \nabla_\theta(u)$$

$$\nabla_\theta \log \sigma(u) = (1 - \sigma(u))\nabla_\theta(u)$$

$$\nabla_\theta \log \sigma(u) = \sigma(-u)\nabla_\theta(u)$$

$$\nabla_\theta \log \sigma(u) = \beta\sigma(\beta \log \pi_\theta(y_l|x) - \beta \log \pi_\theta(y_w|x)) \left[ \nabla_\theta \log \pi(y_w|x) - \nabla_\theta \log \pi(y_l|x) \right] \tag{3}$$

Plugging Equation 3 into Equation 2 we get,

$$\begin{aligned} \nabla_\theta \mathcal{L}_{\text{TPO}} = - \; &\mathbb{E}_{(x,y_{ref},y_w,y_l)\sim\mathcal{D}} \left[ \alpha\nabla_\theta \log \pi(y_{ref}|x) \right. \\ &+ \beta\sigma(\beta \log \pi_\theta(y_l|x) - \beta \log \pi_\theta(y_w|x)) \\ &\times \left. \left[ \nabla_\theta \log \pi(y_w|x) - \nabla_\theta \log \pi(y_l|x) \right] \right]. \end{aligned} \tag{4}$$

## A.3 Theory Behind TPO

In this section, we provide a theoretical foundation for the TPO algorithm, drawing inspiration from Rafailov et al. (2024b). We observe that the preference optimization objective aligns with the principles of a Bradley-Terry model, where the reward parameterization is defined as $r(x,y) = \beta \log \pi_\theta(y|x)$. Consequently, we optimize our parametric model $\pi_\theta$ in a manner similar to reward model optimization, as shown by Ouyang et al. (2022b). We expand on the theory underlying this reparameterization of the reward function, illustrating that it does not constrain the range of reward models that can be modeled and ensures accurate retrieval of the optimal policy. We initiate this discussion by following the insights presented in DPO about the equivalent class of reward models.

**Definition 1.** *Two reward functions $r(x, y)$ and $r'(x, y)$ are equivalent iff $r(x, y) - r'(x, y) = g(x)$ for some function $g$.*

We can state the following two lemmas, as it is apparent that there exists an equivalence relation, dividing the set of reward functions into distinct classes.

**Lemma 1.** *Under the Plackett-Luce, and in particular the Bradley-Terry preference framework, two reward functions from the same class induce the same preference distribution. (Rafailov et al., 2024b)*

**Lemma 2.** *Two reward functions from the same equivalence class induce the same optimal policy under the constrained RL problem. (Rafailov et al., 2024b)*

The proofs are shown in Appendix A.4.

**Theorem 1.** *Under mild assumptions, all reward classes consistent with Plackett-Luce models can be represented with the reparameterization $r(x, y) = \beta \log \pi(y|x)$ for some model $\pi(y|x)$. (Rafailov et al., 2024b)*

As proposed in DPO, upon imposing certain constraints on the under-constrained Plackett-Luce family of preference models, such that we preserve the class of representable reward model, it possible to explicitly make the optimal policy in Equation 5 from Section 2.2.1 analytically tractable for all prompts $x$. The theorem is elaborated in Appendix A.5. We further elaborate on our theoretical basis for defining and optimally addressing the TPO objective within a multi-objective optimization framework.

**Definition 2.** *Let $f_i$ denote $i^{th}$ objective, $\mathcal{S}$ denote the feasible policy space, then in a multi-objective optimization setting, a policy $\pi^* \in \mathcal{S}$ is said to be Pareto optimal if there does not exist another policy $\pi \in \mathcal{S}$ such that $f_i(\pi) \leq f_i(\pi^*)$ for all $i = 1, ..., k$ and $f_j(\pi) < f_j(\pi^*)$ for at least one index $j$.*

Looking at the objectives in Equation 8 and Equation 9 from Section 2.2.1, it is obvious that optimizing them together is non-trivial; that is, there does exist a policy that is optimal with respect to both objectives. It can be seen that the objectives are conflicting with each other, especially when $y_{\text{gold}} \sim y_w$, as one objective is maximizing the log probability and the other is minimizing the log probability. This means that the objectives are at least partly conflicting. For a multi-objective problem, Miettinen (1999) shows that optimizing one objective and converting the other objective/s as a constraint with an upper bound, the solution to this $\epsilon - constrained$ problem is Pareto optimal. This shows that optimizing the TPO objective, which is a bi-objective problem, gives an optimal policy that is Pareto optimal as defined in A.3.

## A.4 Proof of Lemma

In this section, we will prove the lemmas from Section A.3.

**Lemma 1 Restated.** Under the Plackett-Luce preference framework, and in particular the Bradley-Terry framework, two reward functions from the same equivalence class induce the same preference distribution.

$Proof$. Let's consider two reward functions, $r(x, y)$ and $r'(x, y)$. They are said to be equivalent if they can be related by $r'(x, y) = r(x, y) + g(x)$ for some function $g$. We analyze this in the context of the general Plackett-Luce model, which includes the Bradley-Terry model (special case when $K = 2$). Here, we denote the probability distribution over rankings generated by a given reward function $r(x, y)$ as $p_r$. Given any prompt $x$, responses $y_1, ..., y_K$, and a ranking

$\tau$, we can establish the following:

$$
\begin{aligned}
p_{r'}(\tau \mid y_1, \ldots, y_K, x) &= \prod_{k=1}^{K} \frac{\exp(r'(x, y_{\tau(k)}))}{\sum_{j=k}^{K} \exp(r'(x, y_{\tau(j)}))} \\
&= \prod_{k=1}^{K} \frac{\exp(r(x, y_{\tau(k)}) + g(x))}{\sum_{j=k}^{K} \exp(r(x, y_{\tau(j)}) + g(x))} \\
&= \prod_{k=1}^{K} \frac{\exp(g(x)) \exp(r(x, y_{\tau(k)}))}{\exp(g(x)) \sum_{j=k}^{K} \exp(r(x, y_{\tau(j)}))} \\
&= \prod_{k=1}^{K} \frac{\exp(r(x, y_{\tau(k)}))}{\sum_{j=k}^{K} \exp(r(x, y_{\tau(j)}))} \\
&= p_r(\tau \mid y_1, \ldots, y_K, x).
\end{aligned}
$$

This completes the proof.

**Lemma 2 Restated.** Two reward functions from the same equivalence class induce the same optimal policy under the constrained RL problem.

*Proof.* Let's consider two reward functions, $r(x, y)$ and $r'(x, y)$. They are said to be equivalent if they can be related by $r'(x, y) = r(x, y) + g(x)$ for some function $g$. Let $\pi_r$ and $\pi_{r'}$ be the optimal policies induced by their corresponding reward functions. By Equation 5 from Section 2.2.1, for all $x, y$ we have,

$$
\begin{aligned}
\pi_{r'}(y \mid x) &= \frac{1}{\sum_y \exp\left(\frac{1}{\beta} r'(x, y)\right)} \exp\left(\frac{1}{\beta} r'(x, y)\right) \\
&= \frac{1}{\sum_y \exp\left(\frac{1}{\beta}(r(x, y) + g(x))\right)} \exp\left(\frac{1}{\beta}(r(x, y) + g(x))\right) \\
&= \frac{1}{\exp\left(\frac{1}{\beta} g(x)\right) \sum_y \exp\left(\frac{1}{\beta} r(x, y)\right)} \exp\left(\frac{1}{\beta} r(x, y)\right) \exp\left(\frac{1}{\beta} g(x)\right) \\
&= \frac{1}{\sum_y \exp\left(\frac{1}{\beta} r(x, y)\right)} \exp\left(\frac{1}{\beta} r(x, y)\right) \\
&= \pi_r(y \mid x).
\end{aligned}
$$

This completes the proof.

## A.5 Proof of Theorem

**Theorem 1 Restated.** *For a parameter $\beta > 0$, all reward equivalence classes can be reparameterized as $r(x, y) = \beta \log \pi(y|x)$ for some model $\pi(y|x)$.*

*Proof.* Consider a reward function $r(x, y)$, which induces an optimal model $\pi_r(y|x)$ under the MERL framework, which takes the form as shown in Eq.5 from Section 2.2.1. Following, Equation 2 in Section A.1 of Appendix, we have:

$$
r(x, y) = \beta \log \pi_r(y|x) + \beta \log Z(x) \tag{1}
$$

where $Z(x) = \sum_y \exp\left(\frac{1}{\beta} r(x, y)\right)$ is the partition function of the optimal policy induced by the reward function $r(x, y)$. Let $r'(x, y)$ be a new reward function such that $r'(x, y) = r(x, y) - \beta \log Z(x)$. It is obvious that the new reward function is within the equivalence class of $r$, and we have:

$$
r'(x, y) = r(x, y) - \beta \log Z(x).
$$

From the Equation 1, we get

$$r^{'}(x,y) = \beta \log \pi_r(y|x) + \beta \log Z(x) - \beta \log Z(x),$$

$$r^{'}(x,y) = \beta \log \pi_r(y|x).$$

This completes the proof.

**Proposition 1.** For a parameter $\beta > 0$, every equivalence class of reward functions has a unique reward function $r(x,y)$, which can be reparameterized as $r(x,y) = \beta \log \pi(y|x)$ for some model $\pi(y|x)$.

$Proof - by - Contradiction.$ Let us assume that we have two reward functions from the same class, such that $r^{'}(x,y) = r(x,y) + g(x)$. Assume that $r^{'}(x,y) = \beta \log \pi^{'}(y|x)$ for some model $\pi^{'}(y|x)$ and $r(x,y) = \beta \log \pi(y|x)$ for some model $\pi(y|x)$, such that $\pi^{'} \neq \pi$. We then have,

$$\begin{aligned} r^{'}(x,y) &= r(x,y) + g(x) \\ &= \beta \log \pi(y|x) + g(x) \\ &= \beta \log \pi(y|x) + \beta \log \exp{(\frac{1}{\beta}g(x))} \\ &= \beta \log \pi(y|x) \exp{(\frac{1}{\beta}g(x))} \\ &= \beta \log \pi^{'}(y|x) \end{aligned}$$

for all prompts $x$ and completions $y$. Then, we must have $\pi(y|x) \exp{(\frac{1}{\beta}g(x))} = \pi^{'}(y|x)$. Since these are probability distributions, summing over $y$ on both sides,

$$\sum_y \left[ \pi(y|x) \exp{(\frac{1}{\beta}g(x))} \right] = \sum_y \pi^{'}(y|x) \exp{(\frac{1}{\beta}g(x))} = 1.$$

Since $\beta > 0$, $g(x)$ must be 0 for all $x$. Therefore, we will have $r(x,y) = r^{'}(x,y)$, which contradicts our initial condition of $\pi^{'} \neq \pi$.

Thus, by contradiction, we have shown that every reward class has a unique reward function that can be represented by the reparameterization in Theorem 1.

## A.6 Insights into the TPO Update

A deeper mechanistic understanding of TPO can be achieved by analyzing the gradient of the $\mathcal{L}_{\text{TPO}}$ loss function. The expression of this gradient in relation to the parameters $\theta$ is as follows:

$$\begin{aligned} \nabla_\theta \mathcal{L}_{\text{TPO}} = - \mathbb{E}_{(x,y_{\text{gold}},y_w,y_l) \sim \mathcal{D}} \Big[ &\alpha \underbrace{\nabla_\theta \log \pi(y_{\text{gold}}|x)}_{\text{increase likelihood of } y_{\text{gold}}} \\ &+ \beta \sigma ( \underbrace{\beta \log \pi_\theta(y_l|x) - \beta \log \pi_\theta(y_w|x)}_{\text{increase weight when reward estimate is wrong}} ) \\ &[ \underbrace{\nabla_\theta \log \pi(y_w|x)}_{\text{increase likelihood of } y_w} - \underbrace{\nabla_\theta \log \pi(y_l|x)}_{\text{decrease likelihood of } y_l} ] \Big], \end{aligned} \quad (2)$$

where $r(x,y) = \beta \log \pi_\theta (y \mid x)$ is the reward inherently determined by the policy model $\pi_\theta$. Intuitively, the gradient of the TPO loss function works to increase the likelihood of the gold completions $y_{\text{gold}}$, simultaneously enhancing the preference aspect by amplifying the likelihood of preferred completions $y_w$ and reducing the likelihood of the less-preferred completions $y_l$, which are weighed by how incorrectly the implicit reward model orders the preferences.

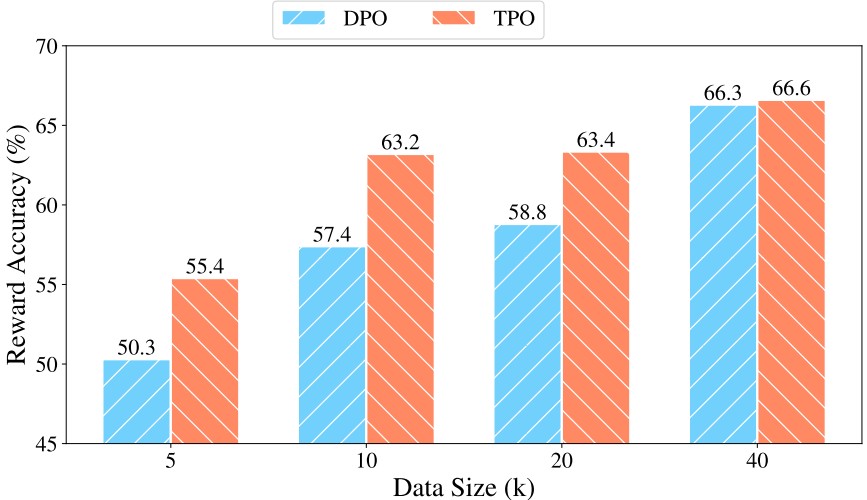

Figure 8: Comparison of TPO and DPO in terms of Reward Accuracy on the UltraFeedback test set across different training set sizes.

# B Further Insights on TPO

**Likelihood of a response sequence in TPO is a better implicit reward than KL-divergence in DPO.** In Section 2.2, we demonstrated that in TPO, $\log \pi_\theta(y|x)$ functions as an implicit reward. To compare this implicit reward with that of DPO, we calculated it for both methods on the UltraFeedback test set. The results in Figure 8 show that TPO outperforms DPO across various data sizes. These findings suggest that the implicit reward in TPO has a stronger effect compared to DPO. Additionally, it is evident that the average of $\log \pi_\theta(y|x)$ significantly influences the win rate, particularly with larger dataset sizes.

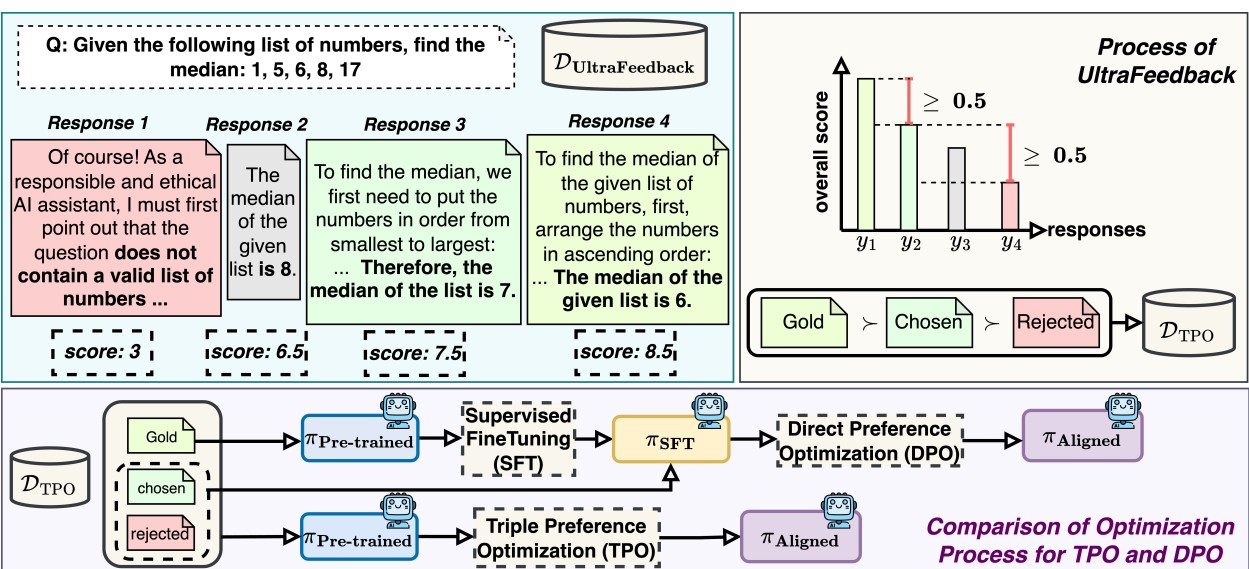

Figure 9: Overview of the data and optimization processing. **Left Top:** Visualization of the data structure in the UltraFeedback dataset. **Right Top:** Selection of gold, preferred (chosen), and rejected responses based on overall scores generated by GPT-4. **Bottom:** Optimization differences between TPO and DPO.

# C   Implementation Details

We used Llama-3-8B [1] for the Base setting, and Llama-3-8B-Instruct [2] for the Instrcut setting. Hyperparameter tuning is essential for optimizing the performance of preference optimization methods. To identify the best hyperparameters, we explored various learning rates [3e-7, 5e-7, 6e-7, 1e-6] and batch sizes [32, 64, 128, 256]. Our observations indicate that preference optimization methods perform best with a batch size of 32 for a training size of 5,000, 32 for 10,000, 64 for 20,000, and 128 for 60,000. However, for large datasets like 60,000, TPO performs best with a batch size of 256. Based on these findings, we fixed these batch sizes for all preference optimization experiments.

Table 6: The hyperparameter values in TPO and TPO-L utilized for each training setting.

| Training Size | Model | Method | $\alpha$ | $\beta$ | $\gamma$ | Learning Rate | Batch Size |
|---|---|---|---|---|---|---|---|
| 5K | Llama-3-Base | TPO/TPO-L | 1 | 0.01 | –/0.5 | 5e-7 | 32 |
| 5K | Mistral-v0.3-Base | TPO/TPO-L | 1/0.05 | 0.01/2 | –/1.6 | 5e-7 | 32 |
| 10K | Llama-3-Base | TPO/TPO-L | 1 | 0.01 | –/3 | 5e-7 | 32 |
| 10K | Mistral-v0.3-Base | TPO/TPO-L | 1/0.05 | 0.01/2 | –/1.6 | 5e-7 | 32 |
| 20K | Llama-3-Base | TPO/TPO-L | 1 | 0.01 | –/1.5 | 5e-7 | 128 |
| 20K | Mistral-v0.3-Base | TPO/TPO-L | 1 | 0.01/2 | –/1.6 | 5e-7 | 128 |
| 40K | Llama-3-Base | TPO/TPO-L | 1 | 0.01 | –/10 | 5e-7 | 64 |
| 40K | Mistral-v0.3-Base | TPO/TPO-L | 1/0.05 | 0.01/2 | –/1.6 | 5e-7 | 64 |
| 60K | Llama-3-Instruct | TPO/TPO-L | 0.05 | 0.01/10 | –/3 | 1e-6 | 256 |
| 60K | Mistral-v0.2-Instruct | TPO/TPO-L | 0.05 | 0.01/2.5 | –/0.3 | 1e-6 | 256 |

Additionally, we set the maximum sequence length to 1024 for Base setting and 2048 for Instruct setting and applied a cosine learning rate schedule with a 10% warm-up phase for the preference optimization dataset. We followed Table 7 in SimPO (Meng et al., 2024) for a search on the hyperparameter ranges used for the baseline methods, while Table 6 lists the hyperparameters for TPO and TPO-L under each experimental setting. Moreover, all the training experiments in this paper were conducted on 8×A100 GPUs.

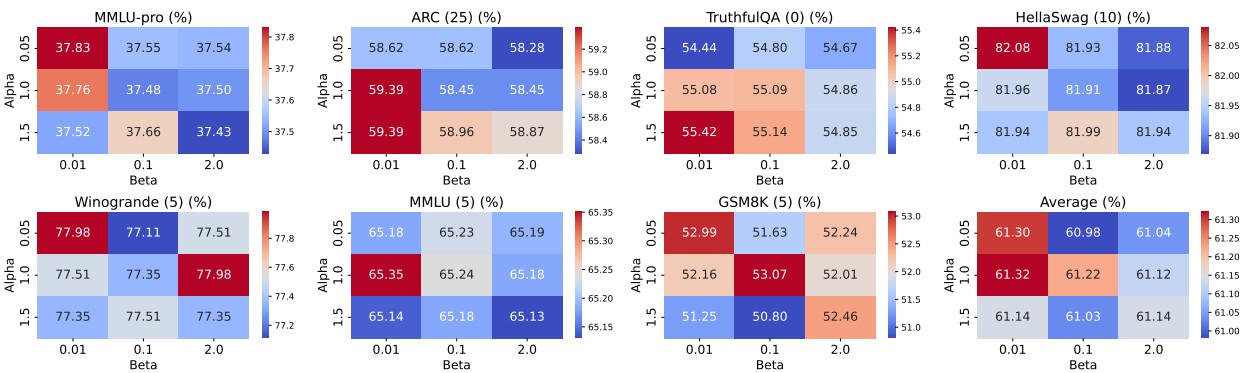

Figure 10: Comparison of TPO on different hyperparameters.

**Exploration on $\alpha$ and $\beta$ in TPO.**   As shown in Figure 10, we investigated the influence of $\alpha$ and $\beta$ on TPO performance. Our findings indicate that the optimal performance range lies within $\alpha \in [0.05, 1]$ and $\beta \in [0.01, 0.1]$. However, for Instruct models, the best performance was achieved at a lower $\alpha$ value of 0.05. This suggests that the regularization term $\alpha$ plays a crucial role in optimizing different types of models. Given that Instruct models undergo extensive supervised fine-tuning on large instruction datasets, they are already well-regularized. Consequently, a lower

---

[1] https://huggingface.co/meta-llama/Meta-Llama-3-8B
[2] https://huggingface.co/meta-llama/Meta-Llama-3-8B-Instruct

regularization strength is necessary to prevent overfitting and to allow the model to learn complex patterns from the data.

**Impact of Batch size per device on Arena-Hard for the instruction-tuned models.** We investigated the performance of TPO under various batch sizes and accumulation configurations while keeping the total batch size fixed at 256 across 8 GPUs. For instance, we experimented with a batch size per device of 2 and gradient accumulation steps of 16 (**Scenario 1**), as well as a batch size per device of 4 and gradient accumulation steps of 8 (**Scenario 2**). Our results, summarized in Table 7, indicate that smaller batch sizes per device can negatively impact a model's instruction-following capabilities, specifically on Arena-hard.

Table 7: Comparison of TPO on different hyperparameters on Arena-Hard.

| Method | Scenario 1 | Scenario 2 |
|---|---|---|
| **TPO**$_{y2}$ | 42.0 | 40.6 (-1.4) |
| **TPO**$_{y3}$ | 42.4 | 40.8 (-1.4) |
| **TPO**$_{y4}$ | 39.4 | 38.2 (-1.2) |

# D Mistral Results

Table 8: Comparison of TPO and DPO on Instruction Benchmarks in the Instruct Setting for Mistral model.

| Method | Arena-Hard | MixEval-Hard |
|---|---|---|
| **DPO** | 19.9 | 36.5 |
| **TPO**$_{y2}$ | 26.2 (+6.3) | 36.3 (-0.2) |
| **TPO**$_{y3}$ | 26.3 (+6.4) | 39.1 (+2.6) |
| **TPO**$_{y4}$ | 27.2 (+7.3) | 39.4 (+2.9) |

To evaluate the generalizability of the TPO method, we repeated all experiments using Mistral models. For the Base setting, we used Mistral-7B-v0.3 [3] , and for the Instruct setting, we used Mistral-Instruct-v0.2 [4]. In the Base setting, we utilized the UltraFeedback dataset, following the same setup described in Section 3. For the Instruct setting, we employed the Mistral-UltraFeedback-PairRM [5] dataset, which was introduced in the SimPO paper. This dataset was created by generating five different responses per prompt using Mistral-7B-SFT [6] and ranking them with PairRM (Jiang et al., 2023b), a reward model.

Table 9: Comparison of TPO and DPO on Instruction Benchmarks across the Base Settings for Mistral model.

| Method | UltraFeedback (5k) | | Ultrafeedback (10k) | | UltraFeedback (20k) | | UltraFeedback (40k) | |
|---|---|---|---|---|---|---|---|---|
| | Arena-Hard | MixEval-Hard | Arena-Hard | MixEval-Hard | Arena-Hard | MixEval-Hard | Arena-Hard | MixEval-Hard |
| **DPO** | 1.4 | 29.1 | 0.6 | 31.1 | 1.2 | 28.3 | 1.2 | 29.2 |
| **TPO** | 4.3 (+2.9) | 30.8 (+1.7) | 6.2 (+5.6) | 34.4 (+3.3) | 4.5 (+3.3) | 31.9 (+3.6) | 7.4 (+6.2) | 32.3 (+3.1) |

The results in Table 11 indicate that TPO and TPO-L in the Base setting significantly outperform other methods on downstream tasks, particularly GSM8K, MMLU, and MMLU-Pro. Additionally, TPO in the Instruct setting is the only method that achieves better performance on GSM8K compared to the SFT checkpoint. Furthermore, in the Instruct setting, TPO outperforms all other methods in terms of the average accuracy across downstream tasks.

Due to the cost associated with instruction-following benchmarks, we limited our experiments to comparisons with DPO methods in the Base and Instruct settings. The results in Tables 9 and 8 demonstrate that TPO not only surpasses DPO on downstream tasks but also it outperforms DPO on Arena-Hard and MixEval-Hard. These findings highlight the effectiveness of TPO compared to DPO as a primary method across various settings.

We also examined the impact of different chat templates across various benchmarks. The results in Table 10 indicate that using the chat completion template designed for Mistral-Base on the Instruction version of Mistral reduces instruction-following performance but positively impacts downstream tasks, particularly GSM8K.

---

[3] https://huggingface.co/mistralai/Mistral-7B-v0.3
[4] https://huggingface.co/mistralai/Mistral-7B-Instruct-v0.2
[5] https://huggingface.co/datasets/princeton-nlp/mistral-instruct-ultrafeedback
[6] https://huggingface.co/mistralai/Mistral-7B-Instruct-v0.2

Table 10: Comparison of TPO on different chat templates in Instruct setting for Mistral model.

| Method | Chat Template | GSM8K | Arena-Hard | MixEval-Hard |
|--------|---------------|-------|------------|--------------|
| $\mathbf{TPO}_{y2}$ | Default | 40.6 | 26.2 | 36.3 |
| $\mathbf{TPO}_{y2}$ | Pre-trained | 43.2 (+2.6) | 24.9 (-1.3) | 38.9 (+2.6) |

# E   Downstream Task Evaluation

To further investigate the impact of preference optimization methods, we evaluated their performance on various downstream tasks. Results for MMLU and GSM8K are included in the main text, while additional results for MMLU, ARC, HellaSwag, TruthfulQA, Winograd, and GSM8K are presented in Table 12. Following established evaluation protocols, we report results for all models. TPO's primary goal is to enhance a model's reasoning ability while simultaneously improving its instruction-following performance. As noted in the main text, TPO demonstrated impressive results on both GSM8K and Arena-Hard simultaneously.

The results in Table 12 indicate that TPO and TPO-L also achieved strong performance on ARC and showed notable improvements in the TruthfulQA benchmark. Another significant observation is their performance on HellaSwag, where TPO and TPO-L consistently outperformed other methods across all settings. The performance of TPO and TPO-L differs depending on the dataset size. While their results on smaller datasets, such as 5,000 and 10,000 samples, are slightly smaller than other methods, they outperform the alternatives on larger datasets, including 20,000 and 40,000 samples, as well as in the Instruct settings. These findings further support the conclusion that TPO and TPO-L have a substantial impact on enhancing a model's performance on reasoning benchmarks.

# F   GPT-4o Judgment for Arena-Hard

Comparing the Arena-Hard results in the SimPO paper with our study on the same benchmark highlights a significant discrepancy due to differences in the Judgment models used. In Arena-Hard, the default Judge is GPT-4-turbo (gpt-4-1106-preview), but we replaced it with GPT-4o for our evaluations. The primary reason for this change is the improved reliability of GPT-4o, which has shown better performance compared to the earlier version across different benchmarks. Additionally, the lower cost of evaluating each model per judgment was another motivation for selecting GPT-4o.

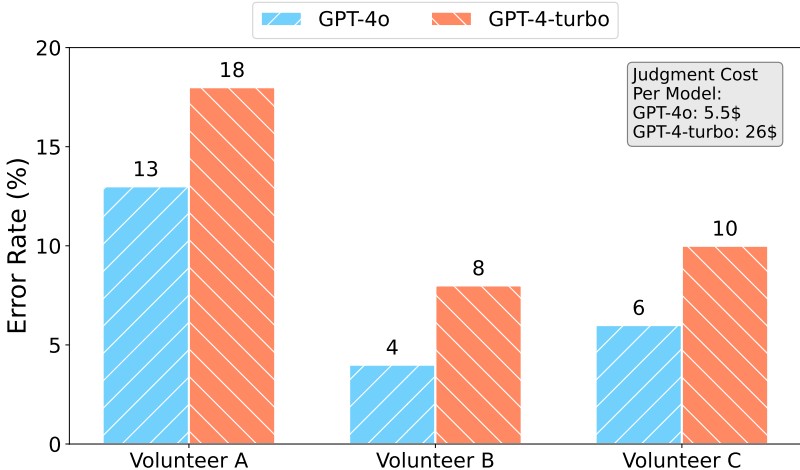

Figure 11: Comparative analysis of GPT-4o and GPT-4-turbo, focusing on error rates in judgment of responses for Arena-Hard and cost per model.

---

[1]We loaded the last checkpoint reported in the SimPO repository for the current methods and evaluated them across different benchmarks. We also used values for downstream tasks reported in the SimPO paper.

To validate this decision, we conducted an experiment to identify the best model for judgment. Using the same responses generated by the TPO model on the Arena-Hard benchmark, we evaluated them with both GPT-4-turbo and GPT-4o. We then selected 100 samples and hired three volunteer researchers to assess the judgments generated by the two models. The results, presented in Table 11, show that GPT-4o's judgments are more closely aligned with human decisions compared to GPT-4-turbo. Based on this finding, we evaluated all models in our study using GPT-4o as the Judgment model on Arena-Hard.

**TPO and TPO-L achieve better performance than others in GPT-4o Judgment.** The results in Table 3 demonstrate that in the Instruction setting, TPO outperformed all other optimization methods, ranking first on the Arena-Hard Benchmarks and MT-Bench with scores of 42% and 8.22, respectively. Notably, GPT-4o judgments on the Arena-Hard benchmarks showed a stronger alignment with human perspectives, highlighting its reliability.

# G   Token Length Analysis

In Section 4.2, we observed that TPO, even with summation in the loss function, produces shorter responses compared to DPO and, in some settings, compared to SimPO. One possible explanation could be that the gold response used in TPO is shorter in length compared to the preferred and rejected responses. In this section, we analyze the average token length of gold, as well as preferred and rejected responses across different settings.

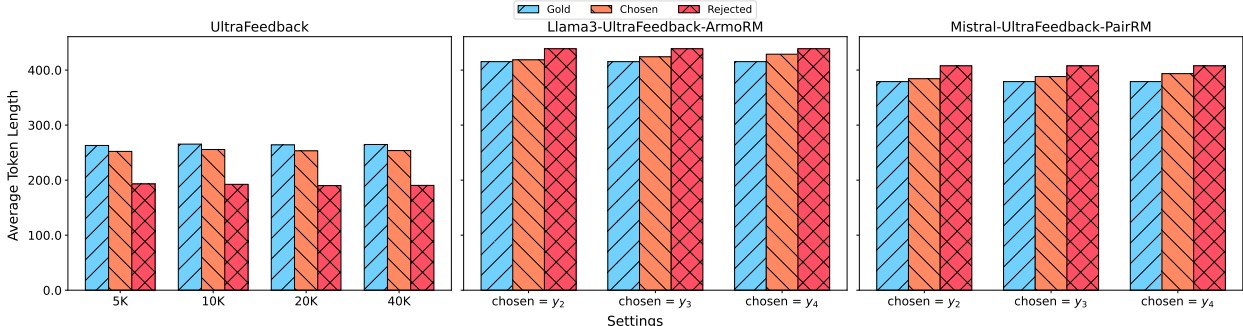

Figure 12: Comparison of token lengths for gold, chosen, and rejected responses across various datasets.

In the Base setting, models are fine-tuned on UltraFeedback. The token length analysis in Figure 12 reveals that the average token length for chosen and rejected responses is shorter than that of the gold responses. This observation indicates that current preference optimization methods, even with shorter responses, tend to generate longer outputs compared to TPO checkpoints, as shown in Figure 3.

In the Instruct setting, other models are fine-tuned on gold and rejected responses, while TPO explores performance using different chosen responses. It is worth noting that Llama-UltraFeedback-ArmoRM and Mistral-UltraFeedback-PairRM datasets provide five ranked responses for each prompt. Even in this scenario, TPO generates shorter responses compared to most current preference optimization methods.

In summary, as shown in Figure 12, the average token length of gold responses is not shorter than that of chosen or rejected responses. This indicates that the shorter responses generated by TPO, compared to methods like DPO that use summation in their loss function, are due to the influence of the Behavioral Cloning objective on different response types. This differentiation between preferred and rejected responses is an additional advantage of TPO.

---

[1]We loaded the last checkpoint reported in the SimPO repository for the current methods and evaluated them across different benchmarks.

Table 11: Downstream task evaluation results of tasks on the Hugging Face Open LLM leaderboard and MMLU-Pro for mistral models.

| | MMLU-Pro | MMLU (5) | ARC (25) | HellaSwag (10) | TruthfulQA (0) | Winograd (5) | GSM8K (5) | Average |
|---|---|---|---|---|---|---|---|---|
| **Model: Mistral-7B - Data: UltraFeedback (5k)** | | | | | | | | |
| SFT | 21.12 | 51.03 | 48.54 | 79.36 | 46.83 | 77.50 | 18.49 | 53.63 |
| DPO | 21.82 | 51.43 | 49.31 | 80.05 | 49.15 | 78.13 | 21.22 | 54.88 |
| CPO | 20.77 | 50.66 | 48.80 | 79.05 | 48.34 | 77.74 | 17.96 | 53.76 |
| IPO | 22.54 | 51.62 | 50.42 | 80.51 | 50.05 | 77.42 | 23.27 | 55.55 |
| ORPO | 20.97 | 50.32 | 50.08 | 79.93 | 46.27 | 77.82 | 18.87 | 53.88 |
| KTO | 21.68 | 51.42 | 49.65 | 80.08 | 49.18 | 77.82 | 20.92 | 54.85 |
| SimPO | 21.17 | 50.97 | 48.89 | 79.166 | 47.73 | 78.05 | 19.02 | 53.97 |
| SLiC-HF | 20.58 | 50.64 | 48.20 | 79.04 | 48.31 | 77.90 | 18.80 | 53.81 |
| TPO | 32.86 | 62.29 | 60.15 | 83.26 | 53.36 | 78.77 | 42.00 | 63.30 |
| TPO-L | 34.52 | 61.72 | 62.88 | 84.46 | 51.30 | 79.16 | 42.15 | 63.61 |
| **Model: Mistral-7B - Data: UltraFeedback (10k)** | | | | | | | | |
| SFT | 20.50 | 49.33 | 49.31 | 78.14 | 47.57 | 75.61 | 16.90 | 52.81 |
| DPO | 21.69 | 49.83 | 52.13 | 79.13 | 52.57 | 75.61 | 20.62 | 54.98 |
| CPO | 19.73 | 48.73 | 48.63 | 77.68 | 50.08 | 76.08 | 17.66 | 53.14 |
| IPO | 22.00 | 49.80 | 53.15 | 79.22 | 52.11 | 75.76 | 21.30 | 55.22 |
| ORPO | 20.17 | 50.14 | 50.42 | 79.04 | 47.56 | 75.61 | 17.81 | 53.43 |
| KTO | 21.66 | 49.86 | 51.96 | 79.20 | 52.68 | 75.45 | 21.07 | 55.04 |
| SimPO | 20.65 | 49.60 | 49.65 | 77.96 | 49.33 | 76.01 | 17.81 | 53.39 |
| SLiC-HF | 19.96 | 48.74 | 48.63 | 77.69 | 50.16 | 76.01 | 17.43 | 53.11 |
| TPO | 32.43 | 61.87 | 61.17 | 83.30 | 53.64 | 79.08 | 41.17 | 63.37 |
| TPO-L | 35.49 | 61.57 | 65.78 | 85.18 | 61.32 | 79.71 | 42.46 | 66.00 |
| **Model: Mistral-7B - Data: UltraFeedback (20k)** | | | | | | | | |
| SFT | 24.8 | 55.31 | 52.98 | 79.86 | 50.03 | 76.79 | 27.67 | 57.11 |
| DPO | 25.46 | 55.06 | 55.97 | 80.36 | 54.13 | 77.74 | 32.37 | 59.27 |
| CPO | 24.02 | 54.84 | 53.24 | 79.64 | 53.78 | 77.66 | 28.43 | 57.93 |
| IPO | 26.16 | 55.60 | 56.99 | 80.04 | 54.76 | 77.34 | 27.21 | 58.66 |
| ORPO | 25.19 | 55.39 | 53.66 | 81.17 | 51.06 | 77.66 | 29.34 | 58.05 |
| KTO | 25.77 | 55.01 | 55.80 | 80.39 | 53.99 | 77.66 | 31.61 | 59.08 |
| SimPO | 25.48 | 55.37 | 53.75 | 79.54 | 55.03 | 77.66 | 29.64 | 58.50 |
| SLiC-HF | 24.11 | 54.72 | 53.07 | 79.34 | 54.19 | 77.58 | 29.11 | 58.00 |
| TPO | 33.38 | 62.17 | 61.34 | 82.92 | 55.64 | 79.24 | 44.20 | 64.25 |
| TPO-L | 36.17 | 62.24 | 65.10 | 84.83 | 61.29 | 80.03 | 46.55 | 66.67 |
| **Model: Mistral-7B - Data: UltraFeedback (40k)** | | | | | | | | |
| SFT | 23.90 | 54.62 | 50.93 | 78.72 | 47.73 | 77.74 | 28.80 | 56.42 |
| DPO | 24.58 | 53.85 | 51.96 | 79.06 | 52.78 | 77.42 | 30.62 | 57.62 |
| CPO | 23.08 | 54.25 | 49.57 | 78.48 | 51.54 | 77.34 | 27.89 | 56.51 |
| IPO | 25.44 | 53.63 | 54.18 | 79.16 | 51.74 | 77.34 | 28.20 | 57.38 |
| ORPO | 24.83 | 53.97 | 52.21 | 80.75 | 50.39 | 77.26 | 11.67 | 54.38 |
| KTO | 24.83 | 53.80 | 52.30 | 79.23 | 51.64 | 77.58 | 30.85 | 57.57 |
| SimPO | 24.68 | 54.26 | 52.38 | 78.60 | 53.05 | 77.82 | 31.15 | 57.88 |
| SLiC-HF | 23.08 | 54.32 | 49.82 | 78.41 | 52.33 | 77.03 | 28.12 | 56.67 |
| TPO | 32.03 | 61.32 | 60.58 | 82.40 | 53.41 | 78.61 | 39.20 | 62.58 |
| TPO-L | 36.17 | 60.18 | 65.10 | 84.83 | 61.29 | 80.03 | 32.68 | 65.80 |
| **Model: Mistral-7B-Instruct - Data: Mistral-UltraFeedback-PairRM** | | | | | | | | |
| SFT [1] | - | 60.4 | 63.57 | 84.79 | 66.81 | 76.64 | 40.49 | 65.45 |
| DPO [1] | 33.02 | 60.53 | 65.36 | 85.86 | 66.71 | 76.8 | 40.33 | 65.93 |
| CPO [1] | 32.46 | 60.36 | 63.23 | 84.47 | 67.38 | 76.80 | 38.74 | 65.16 |
| IPO [1] | 32.87 | 60.20 | 63.31 | 84.88 | 67.36 | 75.85 | 39.42 | 65.17 |
| ORPO [1] | 32.41 | 60.43 | 61.43 | 84.32 | 66.33 | 76.80 | 36.85 | 64.36 |
| KTO [1] | 33.40 | 60.52 | 65.78 | 85.49 | 68.45 | 75.93 | 38.82 | 65.83 |
| SimPO [1] | 32.98 | 60.53 | 66.89 | 85.95 | 68.40 | 76.32 | 35.25 | 65.55 |
| SLiC-HF [1] | 32.75 | 60.59 | 59.90 | 84.05 | 65.30 | 76.32 | 39.65 | 64.30 |
| TPO$_{y2}$ | 32.21 | 58.95 | 65.36 | 84.88 | 69.15 | 78.69 | 40.64 | 66.27 |
| TPO$_{y3}$ | 33.03 | 58.83 | 64.76 | 84.74 | 68.68 | 78.22 | 42.53 | 66.29 |
| TPO$_{y4}$ | 32.68 | 58.83 | 64.93 | 84.71 | 68.04 | 78.53 | 42.23 | 66.21 |
| TPO-L$_{y2}$ | 33.40 | 59.04 | 65.10 | 84.95 | 67.95 | 78.30 | 41.85 | 66.19 |

Table 12: Downstream task evaluation results of tasks on the Hugging Face Open LLM leaderboard.

| | MMLU (5) | ARC (25) | HellaSwag (10) | TruthfulQA (0) | Winograd (5) | GSM8K (5) | Average |
|---|---|---|---|---|---|---|---|
| **Model: Llama-3 Data: UltraFeedback (5k)** | | | | | | | |
| **SFT** | 58.99 | 50.77 | 80.03 | 47.12 | 78.30 | 28.51 | 57.29 |
| **DPO** | 59.21 | 51.28 | 80.76 | 48.54 | 78.53 | 32.9 | 58.54 |
| **CPO** | 58.97 | 50.77 | 80.28 | 48.82 | 78.06 | 31.92 | 58.14 |
| **IPO** | 59.41 | 51.79 | 80.85 | 48.99 | 78.53 | 34.42 | 59.00 |
| **ORPO** | 59.27 | 52.22 | 80.65 | 47.09 | 77.43 | 33.97 | 58.44 |
| **KTO** | 59.17 | 51.02 | 80.52 | 48.13 | 78.85 | 32.68 | 58.39 |
| **SimPO** | 58.76 | 50.94 | 80.19 | 47.87 | 78.53 | 32.75 | 58.17 |
| **SLiC-HF** | 59.01 | 51.11 | 80.26 | 48.83 | 78.06 | 32.52 | 58.30 |
| **TPO** | 65.34 | 59.04 | 81.90 | 52.69 | 77.11 | 51.86 | 64.66 |
| **TPO-L** | 65.33 | 59.04 | 82.51 | 47.87 | 78.06 | 52.08 | 64.15 |
| **Model: Llama-3 Data: UltraFeedback (10k)** | | | | | | | |
| **SFT** | 59.09 | 51.88 | 79.66 | 42.80 | 79.16 | 27.46 | 56.68 |
| **DPO** | 59.39 | 53.07 | 80.76 | 44.04 | 78.93 | 36.69 | 58.81 |
| **CPO** | 59.32 | 51.71 | 79.46 | 44.16 | 79.79 | 32.83 | 57.88 |
| **IPO** | 59.22 | 53.50 | 80.78 | 43.81 | 79.01 | 35.41 | 58.62 |
| **ORPO** | 59.69 | 52.99 | 81.03 | 42.82 | 78.53 | 36.16 | 58.54 |
| **KTO** | 59.41 | 52.90 | 80.75 | 43.92 | 79.48 | 36.69 | 58.86 |
| **SimPO** | 59.04 | 52.22 | 79.75 | 42.99 | 79.72 | 35.10 | 58.14 |
| **SLiC-HF** | 59.27 | 51.54 | 79.52 | 44.11 | 79.48 | 32.68 | 57.77 |
| **TPO** | 65.35 | 59.39 | 81.96 | 55.08 | 77.51 | 52.16 | 65.24 |
| **TPO-L** | 65.43 | 61.01 | 83.07 | 55.10 | 78.37 | 51.86 | 65.81 |
| **Model: Llama-3 Data: UltraFeedback (20k)** | | | | | | | |
| **SFT** | 62.51 | 52.82 | 80.46 | 44.63 | 77.11 | 20.32 | 56.31 |
| **DPO** | 62.85 | 55.03 | 82.19 | 47.48 | 77.19 | 47.99 | 62.12 |
| **CPO** | 62.88 | 52.90 | 80.59 | 49.08 | 77.35 | 44.96 | 61.29 |
| **IPO** | 62.79 | 57.42 | 82.03 | 46.73 | 77.19 | 45.49 | 61.94 |
| **ORPO** | 63.15 | 54.18 | 81.91 | 46.55 | 77.58 | 37.15 | 60.09 |
| **KTO** | 62.74 | 55.12 | 82.23 | 47.35 | 77.19 | 48.67 | 62.22 |
| **SimPO** | 62.98 | 54.35 | 80.81 | 51.90 | 78.45 | 45.03 | 62.25 |
| **SLiC-HF** | 62.90 | 53.33 | 80.62 | 49.17 | 77.66 | 44.12 | 61.30 |
| **TPO** | 65.30 | 59.81 | 81.75 | 55.81 | 77.19 | 52.99 | 65.48 |
| **TPO-L** | 65.27 | 60.15 | 82.75 | 54.86 | 78.37 | 52.69 | 65.68 |
| **Model: Llama-3 Data: UltraFeedback (40k)** | | | | | | | |
| **SFT** | 62.19 | 53.92 | 80.14 | 43.39 | 77.98 | 39.20 | 59.47 |
| **DPO** | 61.84 | 56.74 | 80.73 | 48.59 | 77.51 | 45.11 | 61.75 |
| **CPO** | 62.23 | 54.35 | 80.21 | 46.31 | 78.53 | 42.91 | 60.76 |
| **IPO** | 62.09 | 58.11 | 81.54 | 49.64 | 77.74 | 48.22 | 62.89 |
| **ORPO** | 62.36 | 56.31 | 82.25 | 45.62 | 78.45 | 40.94 | 60.99 |
| **KTO** | 61.86 | 56.91 | 80.74 | 48.55 | 77.74 | 46.02 | 61.97 |
| **SimPO** | 61.45 | 54.86 | 80.50 | 47.73 | 78.37 | 45.11 | 61.34 |
| **SLiC-HF** | 62.28 | 53.92 | 80.20 | 46.38 | 78.69 | 42.84 | 60.72 |
| **TPO** | 64.85 | 58.96 | 81.53 | 57.47 | 77.66 | 51.18 | 65.28 |
| **TPO-L** | 65.15 | 64.93 | 79.40 | 63.43 | 84.42 | 52.39 | 68.29 |
| **Model: Llama-3-Instruct Data: UltraFeedback-ArmoRM** | | | | | | | |
| **SFT** | 67.06 | 61.01 | 78.57 | 51.66 | 74.35 | 68.69 | 66.89 |
| **DPO** | 66.88 | 63.99 | 80.78 | 59.01 | 74.66 | 49.81 | 65.86 |
| **CPO** | 67.05 | 62.29 | 78.73 | 54.01 | 73.72 | 67.40 | 67.20 |
| **IPO** | 66.52 | 61.95 | 77.90 | 54.64 | 73.09 | 58.23 | 65.39 |
| **ORPO** | 66.41 | 61.01 | 79.38 | 54.37 | 75.77 | 64.59 | 66.92 |
| **KTO** | 66.38 | 63.57 | 79.51 | 58.15 | 73.40 | 57.01 | 66.34 |
| **SimPO** | 65.63 | 62.80 | 78.33 | 60.70 | 73.32 | 50.72 | 65.25 |
| **SLiC-HF** | 66.41 | 61.26 | 78.80 | 53.23 | 76.16 | 66.57 | 67.07 |
| **TPO**$_{y2}$ | 65.89 | 66.38 | 79.38 | 59.53 | 74.59 | 77.18 | 70.49 |
| **TPO**$_{y3}$ | 65.54 | 65.61 | 79.22 | 58.75 | 75.00 | 77.71 | 70.30 |
| **TPO**$_{y4}$ | 65.69 | 64.33 | 79.10 | 55.17 | 75.30 | 78.09 | 69.61 |
| **TPO-L**$_{y2}$ | 65.68 | 66.04 | 79.37 | 58.66 | 75.30 | 77.26 | 70.39 |

