# OpenReview forum: "Triple Preference Optimization: Achieving Better Alignment using a Single Step Optimization"
_TMLR — Accepted by TMLR_

### Review · Reviewer_HpRG · 2025-07-21

**Summary Of Contributions:**

This paper introduces Triple Preference Optimization (TPO), a one-step preference optimization method designed to improve both instruction-following and reasoning capabilities of language models. Unlike prior approaches such as DPO and SimPO that rely on two-way preferences, TPO incorporates an additional “gold” response and combines a behavioral cloning (BC) loss with a preference-based objective. The authors also propose TPO-L, a length-controlled variant to reduce verbosity. Extensive experiments on LLaMA and Mistral models across multiple benchmarks show that TPO/TPO-L consistently outperform existing baselines, especially in low-data and noisy settings.

**Audience:**

Yes

**Claims And Evidence:**

No

**Requested Changes:**

Please address the above weaknesses and better distinguish TPO from DPO/SimPO + NLL loss.

**Strengths And Weaknesses:**

### Strengths
- The idea of leveraging three preferences (`gold ≻ preferred ≻ rejected`) to improve preference optimization performance is intuitive and empirically compelling.
- The paper presents solid experimental results on both instruction-following and reasoning benchmarks across varying training sizes, including low-resource scenarios.
- TPO shows notable robustness to label noise and avoids common verbosity issues observed in DPO and its variants.

### Weaknesses
- My primary concern is the limited novelty. The proposed TPO/TPO-L essentially extends DPO/SimPO by adding an NLL regularization term that learns a high-quality (gold) response. However, the idea of combining DPO with an NLL loss has already been explored in prior work (e.g., [1], [2]), and the new insights provided by TPO are fairly limited.

- Additionally, the reliance on high-quality triple preferences restricts the method’s practical applicability. Constructing (`gold`, `preferred`, `rejected`) triplets is significantly more expensive than collecting pairwise preferences, and the paper does not adequately address how scalable or reproducible this process is. The assumption that such fine-grained rankings can be reliably obtained across tasks or domains may not hold in practice.

- Finally, the reward margin hyperparameter $\gamma$ in TPO-L plays a critical role, but its optimal value appears to be task-specific. The tuning process is underexplained, which introduces brittleness for practitioners looking to apply TPO-L out of the box.

### References
[1] Iterative Reasoning Preference Optimization. https://arxiv.org/pdf/2404.19733

[2] The LLaMA 3 Herd of Models. https://arxiv.org/pdf/2407.21783

---

> ### Author Response · Authors · 2025-10-21
> **Official Reply to Reviewer HpRG - Part 1**
>
> We appreciate your careful review and valuable insights.
>
> ---
> > **W1:** My primary concern is the limited novelty. The proposed TPO/TPO-L essentially extends DPO/SimPO by adding an NLL regularization term that learns a high-quality (gold) response. However, the idea of combining DPO with an NLL loss has already been explored in prior work (e.g., [1], [2]), and the new insights provided by TPO are fairly limited. **R1:** Please address better distinguish TPO from DPO/SimPO + NLL loss.
>
> ---
> Thank you for your thoughtful comment. We would like to clarify the distinctions between TPO and related methods, including Iterative Reasoning Preference Optimization (IRPO), DPO, and SimPO.
>
> **Comparison with IRPO:** The objective of IRPO is designed for an iterative learning process, in which the reference model is updated across different training iterations. This approach is conceptually closer to Direct Nash Optimization [1]. Suppose we assume the reference model is fixed and will not update. In that case, the IRPO objective is more closely related to CPO, which combines an NLL loss with preference optimization over response pairs.
>
> Our analysis in section 4.3 reveals that combining a behavioral cloning (BC) term with preference optimization can lead to optimization conflicts. Specifically, while the BC component aims to increase the likelihood of preferred responses, it may inadvertently reduce them during optimization, resulting in inconsistent learning behavior. TPO resolves this issue by reformulating the objective function to ensure that optimization consistently increases the likelihood of preferred responses, thereby achieving more stable and effective training dynamics.
>
> **Comparison with DPO:** TPO also differs fundamentally from DPO in both its objective formulation and its dependence on a reference model. DPO relies on a fixed reference model, where the KL-divergence term serves as an implicit reward signal. This dependency constrains optimization, as the policy must balance preference learning with regularization toward the reference model.
>
> In contrast, TPO removes this dependency by directly optimizing the model’s log-likelihood over triplet preferences without referencing a separate model. This formulation allows the policy to focus purely on preference alignment while maintaining stability through its integrated learning objective. As shown in Figure 8 and discussed in Section A.3, our theoretical analysis demonstrates that TPO’s log-probability term provides a stronger and more stable implicit reward representation than that of DPO.
>
> **Comparison with SimPO:** TPO establishes a reference-free preference optimization framework grounded in Maximum Entropy Reinforcement Learning (MERL), combining NLL with preference optimization over triplet preferences. This structure provides a principled theoretical foundation for reference-free learning.
>
> By contrast, SimPO removes the reference model primarily through empirical reasoning rather than a formal theoretical framework. It computes averages during the log-probability calculation, while TPO employs summation, a formulation that preserves stronger probabilistic consistency and leads to more stable optimization. This theoretical grounding, supported by our analysis in Section A.3 and illustrated in Figures 3 and 5, explains why TPO not only produces shorter and more concise responses than SimPO and DPO but also yields a more robust and stable implicit reward signal, reinforcing both its learning stability and overall effectiveness.
>
> Additionally, both SimPO and DPO require an auxiliary optimization step involving a supervised checkpoint, whereas TPO achieves comparable or superior performance without relying on a supervised backbone. Consequently, TPO offers both greater training efficiency and stronger theoretical grounding.
>
> We will incorporate these clarifications in the camera-ready version.
>
> ---
>
> [1] Direct Nash Optimization: https://arxiv.org/abs/2404.03715

---

> ### Author Response · Authors · 2025-10-21
> **Official Reply to Reviewer HpRG - Part 2**
>
> > **R2:** Additionally, the reliance on high-quality triple preferences restricts the method’s practical applicability. Constructing (gold, preferred, rejected) triplets is significantly more expensive than collecting pairwise preferences, and the paper does not adequately address how scalable or reproducible this process is. The assumption that such fine-grained rankings can be reliably obtained across tasks or domains may not hold in practice.
>
> ---
> Thank you for your comment. We provided a detailed explanation of the dataset construction process in Section 2.2 and illustrated it in Figure 9 of Appendix B. We explored the performance of TPO on two different types of datasets, as described below.
>
> **UltraFeedback:** The UltraFeedback dataset was constructed by collecting multiple responses for each prompt from a variety of models. These responses were then evaluated and ranked according to five defined criteria measured by GPT-4. The results presented in Table 2 are based on this dataset and show that both TPO and TPO-L significantly outperform other methods across six benchmarks and four different data scales. For additional details regarding this dataset, please refer to [1].
>
> **UltraFeedback-ArmoRM and UltraFeedback-PairRM:** These datasets were introduced in the SimPO paper as extended versions of UltraFeedback that use reward-based ranking instead of direct GPT-4 scoring. In UltraFeedback-ArmoRM, a LLaMA-8B model was first fine-tuned on the UltraChat dataset, which consists of 200K high-quality conversations generated by ChatGPT [2]. For each prompt, five diverse responses were then generated by the fine-tuned model, and the ArmoRM reward model was applied to assign scores and determine their ranking. The results reported in Table 3 correspond to this dataset, where both TPO and TPO-L achieve strong and consistent improvements.
>
>
> Similarly, in UltraFeedback-PairRM, the authors followed the same procedure using a Mistral model, ranking the generated responses with the PairRM reward model. The results for these Mistral-based models are included in Appendix D. For more details about these datasets, please refer to the SimPO paper.
>
> These examples demonstrate that constructing a preference dataset is not computationally expensive and follows a process similar to that of building pairwise preference datasets. In fact, creating a pairwise dataset also requires generating multiple responses, ranking them, and selecting the best pair. In the case of TPO, we only need to select one additional response to serve as the gold response, which minimally increases the curation cost.
>
> Moreover, the robustness experiments presented in Section 4 show that TPO is less sensitive to noise in preference data. During the rebuttal, we also analyzed the effect of noise in the gold response, as detailed in our response to Reviewer HCqd.
> In conclusion, TPO is not only robust to noise in preference data but also cost-efficient in terms of data curation, requiring a comparable level of effort to other pairwise preference optimization methods.
>
> ---
> > **R3:** Finally, the reward margin hyperparameter 𝛾  in TPO-L plays a critical role, but its optimal value appears to be task-specific. The tuning process is underexplained, which introduces brittleness for practitioners looking to apply TPO-L out of the box.
>
> ---
>
> Thank you for your comment. We agree with the reviewer that gamma is a task-specific hyperparameter, and its suitable value can vary depending on the dataset size and characteristics. To determine the most suitable value for gamma, we conducted a linear search starting from 0.5 and gradually increasing up to approximately [1, 1.5]. This process continued until the model achieved the highest performance relative to other methods. The complete set of exploration results is presented in Figure 6, and the suitable gamma values for the other datasets are reported in Table 6 in Appendix C.
>
> We also acknowledge that identifying the optimal gamma value remains an open research problem. Given its importance, we consider further exploration of this aspect an interesting direction for future work. We will revise Appendix C to include a more detailed explanation of the gamma selection process for TPO-L.
>
>
> ---
>
> [1] UltraFeedback: https://huggingface.co/datasets/openbmb/UltraFeedback
>
> [2] UltraChat: https://huggingface.co/datasets/HuggingFaceH4/ultrachat_200k

---

### Review · Reviewer_Ma7b · 2025-09-08

**Summary Of Contributions:**

The paper introduces Triple Preference Optimization (TPO), a novel single-step preference learning method designed to overcome key limitations of Direct Preference Optimization (DPO). Unlike existing approaches that often suffer from optimization inefficiency, poor reasoning performance, sensitivity to dataset size, and vulnerability to noisy judgments, TPO integrates behavioral cloning on gold responses with a reference-free preference objective across three levels of preference. The authors also propose a length-controlled variant, TPO-L, to mitigate verbosity issues. They provide a theoretical derivation and empirical experiments on both base and instruction-tuned models. The result shows that TPO and TPO-L consistently outperform DPO and its variants across a wide range of instruction-following and reasoning benchmarks, achieve robustness to noisy preference data, and deliver superior performance even with reduced training data.

**Audience:**

Yes

**Claims And Evidence:**

No

**Requested Changes:**

The paper would benefit from a clearer discussion of its assumptions and positioning, especially the reliance on triple-preference data. Moreover, while the empirical evaluation is strong, the paper should situate TPO more explicitly within the broader baseline by experimenting with or discussing related approaches such as GRPO, ReMax.

**Strengths And Weaknesses:**

**Strengths:**
The paper proposes Triple Preference Optimization (TPO), a novel single-step preference learning method that effectively addresses key limitations of DPO and its variants. The approach is theoretically grounded in Maximum Entropy RL and preference modeling, and the introduction of TPO-L provides a practical solution to verbosity control. The experiments are extensive, covering multiple models, dataset sizes, and both instruction-following and reasoning benchmarks, and consistently show significant improvements in performance, robustness to noise, and data efficiency.

**Weaknesses:**
The method assumes access to triple-preference data, which may be difficult to obtain in many real-world RLHF settings and raises questions about scalability. Moreover, while comparisons to many strong baselines are provided, the paper would benefit from a clearer discussion of methods like GRPO to better situate TPO within the broader preference optimization landscape and clarify domain applicability.

---

> ### Author Response · Authors · 2025-10-21
> **Official Reply to Reviewer Ma7b**
>
> We sincerely appreciate your thoughtful review and insightful feedback.
>
> ---
> > **W1:** The method assumes access to triple-preference data, which may be difficult to obtain in many real-world RLHF settings and raises questions about scalability. **R1:** The paper would benefit from a clearer discussion of its assumptions and positioning, especially the reliance on triple-preference data.
>
> ---
>
> Thank you for the insightful comment. In general, there are two common approaches for curating preference datasets:
>
> **1. Distillation-based method:** In this approach, multiple open-source or closed-source models are used to generate diverse responses for each prompt. These responses are then ranked by a judger model, typically a strong frontier model, based on a specific evaluation criterion. The UltraFeedback dataset is an example of this method. Refer to [1]  for more details.
>
> **2. Sampling from a supervised fine-tuned (SFT) model:** In this approach, a model is first fine-tuned using SFT data, and then multiple responses are generated for each prompt using a temperature greater than zero. These responses are subsequently ranked by a reward model or a judger model. This method was explored in the SimPO paper, which introduced the UltraFeedback-PairRM and UltraFeedback-ArmoRM datasets.
>
> In our work, we utilized all three datasets, UltraFeedback, UltraFeedback-PairRM, and UltraFeedback-ArmoRM. These approaches effectively address the challenges associated with constructing high-quality preference datasets. We emphasize that even when creating pairwise preference datasets, more than two responses must typically be generated; therefore, using three responses is both feasible and justified.
>
> Additionally, the robustness experiments presented in Section 4.4 demonstrate that TPO is less sensitive to variations in the preference data. Even when two responses with identical evaluation scores are used as the gold and chosen responses, TPO still achieves superior performance compared to the DPO method. Please refer to Table 4 in our paper for further details. We will include this clarification in the camera-ready version.
>
> ---
> > **W2:** Moreover, while comparisons to many strong baselines are provided, the paper would benefit from a clearer discussion of methods like GRPO to better situate TPO within the broader preference optimization landscape and clarify domain applicability. **R2:** Moreover, while the empirical evaluation is strong, the paper should situate TPO more explicitly within the broader baseline by experimenting with or discussing related approaches such as GRPO and ReMax.
> ---
>
> Thank you for the comment. We would like to clarify that TPO is an offline optimization method; therefore, we compared it only with existing state-of-the-art offline preference optimization approaches. In contrast, methods such as GRPO, REINFORCE, RLHF, and ReMax are online approaches, where a policy model is optimized on responses generated during training based on feedback from a reward model or reward function.
>
> We have left the investigation of TPO’s behavior and effectiveness in an online learning setting for future work, as noted in the Limitations and Future Work section. Similar to the clarification provided in the ReMax paper specifically, *“We note that ReMax and DPO are totally different optimization methods,”* and *“ReMax employs an online learning paradigm, whereas DPO updates its models offline”*, we will include a paragraph in the camera-ready version to clearly distinguish TPO from GRPO, ReMax, and other online or iterative optimization methods.
>
> ---
>
> [1] UltraFeedback: https://huggingface.co/datasets/openbmb/UltraFeedback

---

### Review · Reviewer_HCqd · 2025-10-07

**Summary Of Contributions:**

This paper introduces Triple Preference Optimization (TPO), a one-step alignment method for large language models that incorporates three preference levels (gold > preferred > rejected). By combining behavioral cloning on gold responses with preference optimization between preferred and rejected responses, TPO and its length-controlled variant TPO-L achieve stronger performance than DPO and recent variants across both reasoning and instruction-following benchmarks, with improved robustness to data size and noise.

**Audience:**

Yes

**Broader Impact Concerns:**

None noted.

**Claims And Evidence:**

Yes

**Requested Changes:**

- I would suggest the authors run controlled ablations where y_gold itself is corrupted, with different corruption rates, to evaluate whether TPO’s advantage persists.

- I suggest the authors conduct gamma sweeps across a broader range of reasoning and instruction datasets to confirm whether gamma indeed governs this trade-off or if the effect is benchmark-dependent.

**Strengths And Weaknesses:**

strengths:

- an interesting method with a fairly principled design
- the method operates in a single optimization step without requiring a reference model.
- the method achieves strong results with substantially less data than DPO.

weakness

- Robustness experiments assume gold responses are always clean while only corrupting pairwise labels. Since TPO depends heavily on the gold-response behavioral cloning term, its reliability under noisy or mis-specified gold annotations remains unverified.

- The analysis of gamma (reward margin in TPO-L) suggests a trade-off between reasoning (improves with higher gamma) and instruction-following (degrades with higher gamma). However, these results are demonstrated on a limited set of benchmarks (notably GSM8K and Arena-Hard). It is unclear whether the observed trade-off reflects a general principle or dataset-specific artifacts. For example, the GSM8K behavior can be because reasoning, or can also be because some characteristics of GSM8K itself (instead of reasoning).

---

> ### Author Response · Authors · 2025-10-21
> **Official Reply to Reviewer HCqd**
>
> Thank you for your constructive feedback regarding our paper. We appreciate the opportunity to enhance our work based on your insightful suggestions.
>
> ---
> >**W1:** Robustness experiments assume gold responses are always clean while only corrupting pairwise labels. Since TPO depends heavily on the gold-response behavioral cloning term, its reliability under noisy or mis-specified gold annotations remains unverified. **R1:** I would suggest the authors run controlled ablations where y_gold itself is corrupted, with different corruption rates, to evaluate whether TPO’s advantage persists.
>
> ---
> **A1:** Following the reviewer’s suggestion, we conducted an additional experiment where the gold response was intentionally assigned a lower score than the chosen or rejected responses. In this context, we define noise as the situation in which the gold response is ranked below both the chosen and rejected responses.
>
> For this experiment, we selected 40,000 samples from the UltraFeedback dataset and extracted three distinct responses per sample, satisfying the following conditions:
>
> $$
> \Delta(S(y_1), S(y_2)) \ge 0.5 \quad \text{and} \quad \Delta(S(y_2), S(y_3)) \ge 0.5 \quad \text{and} \quad y_1 \succ y_2 \succ y_3
> $$
>
> where S(y) denotes the response score measured by GPT-4 on five different criteria, refer to [1] for more information about the UltraFeedback.
>
> We considered two noisy scenarios:
>
> **1. Scenario 1:** y₂ is the gold response, y₁ is the chosen response, and y₃ is the rejected response.
>
> **2. Scenario 2:** y₃ is the gold response, y₁ is the chosen response, and y₂ is the rejected response.
>
> In both cases, the gold responses were intentionally defined as noisy. We then fine-tuned the LLaMA3-8B model using both TPO and DPO approaches under these conditions.
>
> **Table 1**
>
> |Method|MMLU-Pro|GSM8k|ArenaHard|MixEval-Hard|
> |-|:-:|:-:|:-:|:-:|
> |SFT-DPO (wo noise)|33.5|42.9|5.3|25.4|
> |SFT-DPO (Scenario 1)|33.4|37.4|2.0|29.7|
> |SFT-DPO (Scenario 2)|26.0|34.1|0.6|28.1|
> |TPO (wo noise)|37.4|**51.2**|**6.9**|**32.9**|
> |TPO (Scenario 1)|**38.5**|51.1|6.2|31.2|
> |TPO (Scenario 2)|33.8|45.2|5.8|30.85|
>
>
> The results in Table 1 show that TPO checkpoints from both Scenario 1 and Scenario 2 generally achieve slightly lower performance across most benchmarks compared to TPO checkpoints trained on clean data. However, this performance gap remains relatively small. In contrast, the results for DPO checkpoints reveal that the quality of the SFT data plays a crucial role in their effectiveness. Even when the preference data is clean, using a weaker SFT model can cause a significant decline in DPO performance on reasoning and instruction-following benchmarks, such as GSM8K and ArenaHard, with drops of 8.8% and 4.7%, respectively. Overall, these results demonstrate the superior robustness of TPO relative to the DPO algorithm.
>
>
>
> ---
> > **W2:** The analysis of gamma (reward margin in TPO-L) suggests a trade-off between reasoning (improves with higher gamma) and instruction-following (degrades with higher gamma). However, these results are demonstrated on a limited set of benchmarks (notably GSM8K and Arena-Hard). It is unclear whether the observed trade-off reflects a general principle or dataset-specific artifacts. For example, the GSM8K behavior can be because reasoning, or can also be because some characteristics of GSM8K itself (instead of reasoning). **R2:** I suggest the authors conduct gamma sweeps across a broader range of reasoning and instruction datasets to confirm whether gamma indeed governs this trade-off or if the effect is benchmark-dependent.
>
> ---
> **A2:** Thank you for raising this point. In Figure 6 of our paper, we examined the impact of different gamma values on the ArenaHard, MMLU-Pro, GSM8K, and MixEval-Hard benchmarks. The comparison between reasoning and instruction-following benchmarks revealed a clear trade-off. To further demonstrate this effect, we also report results for additional benchmarks, including MMLU, ARC, TruthfulQA, HellaSwag, and Winogrande.
>
> **Table 2**
> |Gamma |ARC |TruthfulQA |HellaSwag |Winogrande |MMLU |GSM8K |AVG|
> |-|-|-|-|-|-|-|-|
> |0.5|0.6109|0.6364|0.8267|0.7806|0.6458|0.4913|0.6658|
> |1.5|0.6177|0.6383|0.8292|0.7822|0.6455|0.4943|0.6678|
> |3|0.6220|0.6419|0.8336|0.7837|0.6466|0.4814|0.6682|
> |4.5|0.6305|0.6425|0.8371|0.7869|0.6465|0.4890|0.6720|
> |5.4|0.6288|0.6469|0.8383|0.7885|0.6481|0.4867|0.6728|
> |7|0.6382|0.6357|0.8408|0.7916|0.6480|0.5110|0.6775|
> |8.5|0.6502|0.6454|0.8426|0.7869|0.6493|0.5049|0.6798|
> |10|0.6493|0.6343|0.8442|0.7940|0.6515|0.5239|0.6828|
> |15|0.6365|0.6073|0.8395|0.7845|0.6497|0.5633|0.6801|
>
> The results in Table 2 indicate that as the gamma value increases, the average performance on traditional benchmarks improves; however, beyond a certain threshold, the performance on instruction-following benchmarks begins to decline.
>
> ---
>
> [1] UltraFeedback: https://huggingface.co/datasets/openbmb/UltraFeedback
>
> ---

---

### Decision · Action_Editor_mWvi · 2025-11-12

**Recommendation:** Accept as is

**Audience:**

Yes

**Audience Explanation:**

The topic of preference optimization for large language model alignment is timely and relevant to the TMLR audience.

**Claims And Evidence:**

Yes

**Claims Explanation:**

The claims are generally well supported by comprehensive experiments and additional analyses provided during rebuttal, including robustness and hyperparameter studies.